# Human genetic analyses of organelles highlight the nucleus in age-related trait heritability

**Rahul Gupta[1,2,3]\*, Konrad J Karczewski[2,3], Daniel Howrigan[2,3], Benjamin M Neale[2,3]\*, Vamsi K Mootha[1,2]\***

[1]Howard Hughes Medical Institute and Department of Molecular Biology, Massachusetts General Hospital, Boston, United States; [2]Broad Institute of MIT and Harvard, Cambridge, United States; [3]Analytic and Translational Genetics Unit, Center for Genomic Medicine, Massachusetts General Hospital, Boston, United States

**Abstract** Most age-related human diseases are accompanied by a decline in cellular organelle integrity, including impaired lysosomal proteostasis and defective mitochondrial oxidative phosphorylation. An open question, however, is the degree to which inherited variation in or near genes encoding each organelle contributes to age-related disease pathogenesis. Here, we evaluate if genetic loci encoding organelle proteomes confer greater-than-expected age-related disease risk. As mitochondrial dysfunction is a 'hallmark' of aging, we begin by assessing nuclear and mitochondrial DNA loci near genes encoding the mitochondrial proteome and surprisingly observe a lack of enrichment across 24 age-related traits. Within nine other organelles, we find no enrichment with one exception: the nucleus, where enrichment emanates from nuclear transcription factors. In agreement, we find that genes encoding several organelles tend to be 'haplosufficient,' while we observe strong purifying selection against heterozygous protein-truncating variants impacting the nucleus. Our work identifies common variation near transcription factors as having outsize influence on age-related trait risk, motivating future efforts to determine if and how this inherited variation then contributes to observed age-related organelle deterioration.

**\*For correspondence:**
rahul_gupta@hms.harvard.edu (RG);
bneale@broadinstitute.org (BMN);
vamsi_mootha@hms.harvard.edu (VKM)

## Introduction

The global burden of age-related diseases such as type 2 diabetes (T2D), Parkinson's disease (PD), and cardiovascular disease (CVD) has been steadily rising due in part to a progressively aging population. These diseases are often highly heritable: for example, narrow-sense heritabilities were recently estimated as 56% for T2D, 46% for general hypertension, and 41% for atherosclerosis (*Wang et al., 2017*). Genome-wide association studies (GWAS) have led to the discovery of thousands of robust associations with common genetic variants (*Claussnitzer et al., 2020*), implicating a complex genetic architecture as underlying much of the heritable risk. These loci hold the potential to reveal underlying mechanisms of disease and spotlight targetable pathways.

Aging has been associated with dysfunction in many cellular organelles (*López-Otín et al., 2013*). Dysregulation of autophagic proteostasis, for which the lysosome is central, has been implicated in myriad age-related disorders including neurodegeneration, heart disease, and aging itself (*Mizushima et al., 2008*), and mouse models deficient for autophagy in the central nervous system show neurodegeneration (*Hara et al., 2006*; *Komatsu et al., 2006*). Endoplasmic reticular (ER) stress has been invoked as central to metabolic syndrome and insulin resistance in T2D (*Ozcan et al., 2004*). Disruption in the nucleus through increased gene regulatory noise from epigenetic alterations (*López-Otín et al., 2013*) and elevated nuclear envelope 'leakiness'

**eLife digest** Getting older increases our risk of experiencing a wide range of diseases, such as diabetes, heart disease and neurodegenerative disease. The genetic variations that we inherit from our parents play a major role in predicting this risk. However, the biological networks involved in this process are extremely complex and remain challenging to decipher.

Prior studies have suggested that specialised structures inside our body's cells, called organelles, may have an important role to play in aging. Organelles represent self-contained biological factories inside each cell, designed to perform specific tasks. Examples include the nucleus, which harbours most of the cell's genetic material, and mitochondria, which help provide cells with energy.

Organelles tend to deteriorate and become dysfunctional with age, and mitochondria in particular are badly affected by the ageing process. A decline in organelle activity has been thought to explain ageing and the development of age-related diseases. However, this has never been systematically tested on a large scale at the inherited genetic level.

Gupta et al. assessed whether common inherited genetic variation in genes associated with ten different organelles could affect the risk of age-related disease, using a database of DNA samples from more than 300,000 individuals. They considered 24 diseases and traits that become more common with advanced age.

Gupta et al. discovered that inherited variants in or near genes associated with the nucleus were consistently linked to age-related disease risks. Most of this signal arose from genes encoding the nuclear transcription factors, proteins that help to control the rate at which genes are expressed. However, variants in genes associated with other organelles, including mitochondria, did not appear to be linked to age-related diseases.

This research suggests that inherited variation in transcription factors in the nucleus could act as genetic levers that increase the risk of common, age-related diseases. It also suggests that common genetic variation in other cellular organelles may not be as heavily involved in the development of such diseases. Such insights into the cellular structures and biological pathways involved in ageing and age-related disease also establish new targets for drugs to prevent or treat disease.

(*D'Angelo et al., 2009*) has been implicated in aging. Dysfunction in the mitochondria has even been invoked as a 'hallmark' of aging (*López-Otín et al., 2013*) and has been observed in many common age-associated diseases (*Lane et al., 2015*; *Petersen et al., 2004*; *Mootha et al., 2003*; *Schapira et al., 1990*; *Bender et al., 2006*; *Wanagat et al., 2001*; *Ashar et al., 2017*). In particular, deficits in mitochondrial oxidative phosphorylation (OXPHOS) have been documented in aging and age-related diseases as evidenced by in vivo (*Estrada et al., 2012*) P-NMR measures (*Petersen et al., 2004*; *Fleischman et al., 2010*), enzymatic activity (*Mootha et al., 2003*; *Schapira et al., 1990*; *Fannin et al., 1999*; *Trounce et al., 1989*; *Kelley et al., 2002*; *Patti et al., 2003*; *Stump et al., 2003*) in biopsy material, accumulation of somatic mitochondrial DNA (mtDNA) mutations (*Bender et al., 2006*; *Wanagat et al., 2001*; *Taylor et al., 2003*), and a decline in mtDNA copy number (mtCN) (*Ashar et al., 2017*).

Given that a decline in organelle function is observed in age-related disease, a natural question is whether inherited variation in loci encoding organelles is enriched for age-related disease risk. Although it has long been known that recessive mutations leading to defects within many cellular organelles can lead to inherited syndromes (e.g. mutations in >300 nuclear DNA (nucDNA)-encoded mitochondrial genes lead to inborn mitochondrial disease; *Frazier et al., 2019*), it is unknown how this extends to common disease. In the present study, we use a human genetics approach to assess common variation in loci relevant to the function of ten cellular organelles. We begin with a deliberate focus on mitochondria given the depth of literature linking it to age-related disease, interrogating both nucDNA and mtDNA loci that contribute to the organelle's proteome. This genetic approach is supported by the observation that heritability estimates of measures of mitochondrial function are substantial (33–65%; *Curran et al., 2007*; *Xing et al., 2008*). We then extend our analyses to nine additional organelles.

To our surprise, we find no evidence of enrichment for genome-wide association signal in or near mitochondrial genes across any of our analyses. Further, of 10 tested organelles, only the nucleus

shows enrichment among many age-associated traits, with the signal emanating primarily from the transcription factors (TFs). Further analysis shows that genes encoding the mitochondrial proteome tend to be tolerant to heterozygous predicted loss-of-function (pLoF) variation and thus are surprisingly 'haplosufficient' – that is, show little fitness cost with heterozygous pLoF. In contrast, nuclear TFs are especially sensitive to gene dosage and are often 'haploinsufficient,' showing substantial purifying selection against heterozygous pLoF. Thus, our work highlights inherited variation influencing gene-regulatory pathways, rather than organelle physiology, in the inherited risk of common age-associated diseases.

## Results

### Age-related diseases and traits show diverse genetic architectures

To systematically define age-related diseases, we turned to recently published epidemiological data from the United Kingdom (U.K.) (*Kuan et al., 2019*) in order to match U.K. Biobank (UKB) (*Sudlow et al., 2015*) cohort. We prioritized traits whose prevalence increased as a function of age (Materials and methods) and were represented in UKB (https://github.com/Nealelab/UK_Biobank_GWAS) and/or had available published GWAS meta-analyses (*Teslovich et al., 2010*; *Ehret et al., 2011*; *Manning et al., 2012*; *Morris et al., 2012*; *Schunkert et al., 2011*; *Estrada et al., 2012*; *Christophersen et al., 2017*; *Pattaro et al., 2016*; *Nalls et al., 2019*; *Lambert et al., 2013*; *Figure 1A*, Appendix 1). We used SNP-heritability estimates from stratified linkage disequilibrium score regression (S-LDSC, https://github.com/bulik/ldsc) (*Finucane et al., 2015*) to ensure that our selected traits were sufficiently heritable (*Supplementary file 1*, Materials and methods, Appendix 1), observing heritabilities across UKB and meta-analysis traits as high as 0.28 (bone mineral density), all with heritability Z-score > 4. We then computed pairwise genetic and phenotypic correlations

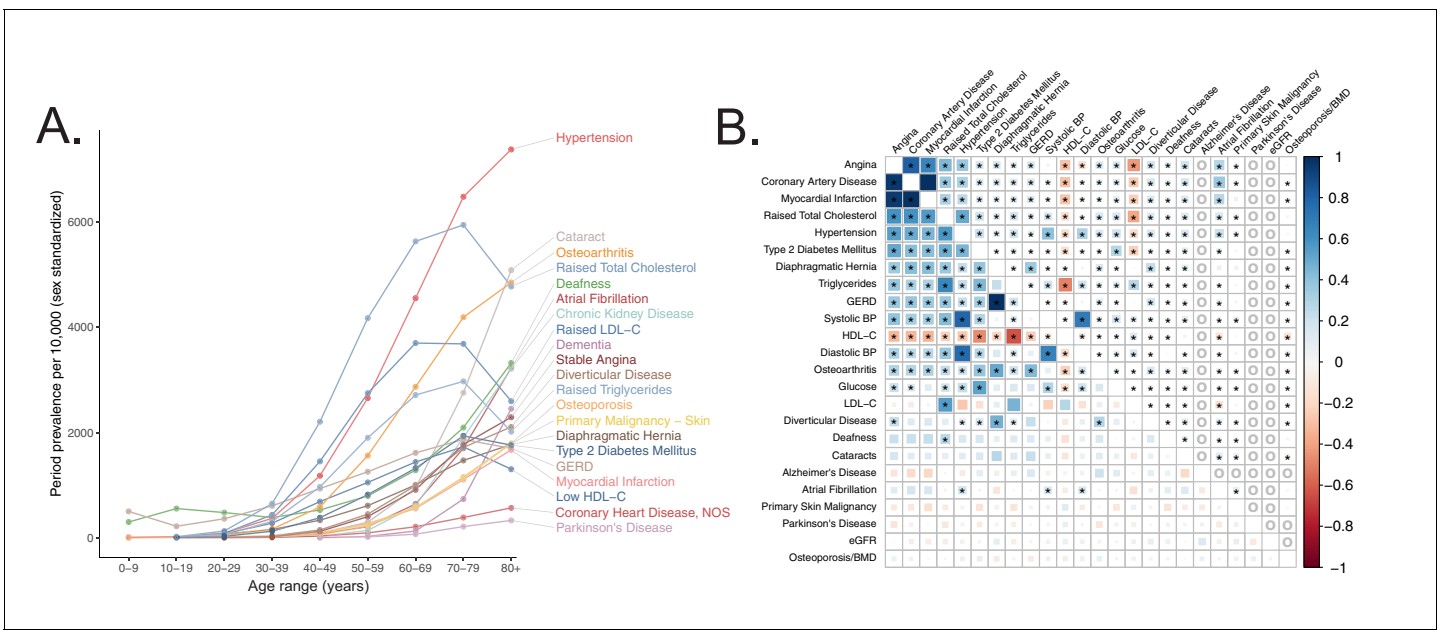

**Figure 1.** Selection of genetically diverse age-related diseases and traits using epidemiological data. (**A**) Period prevalence of age-associated diseases systematically selected for this study (Materials and methods). Epidemiological data obtained from *Kuan et al., 2019*. (**B**) Genetic (lower half) and phenotypic (upper half) correlation between the selected age-related traits. All correlations were assessed between UK Biobank phenotypes with the exception of eGFR, Alzheimer's Disease, and Parkinson's Disease, for which the respective meta-analyses were used (Materials and methods). Grey 'o' in phenotypic correlations indicate phenotypes not tested within UKB for which individual-level data was not available. All data displayed in this panel are available in *Figure 1—source data 1*. * represents correlations that are significantly different from 0 at a Bonferroni-corrected threshold for p = 0.05 across all tested traits.

The online version of this article includes the following source data for figure 1:

**Source data 1.** Genetic and phenotypic correlation point estimates and standard errors.

between the age-associated traits to compare their respective genetic architectures and phenotypic relationships (*Figure 1B*, Materials and methods). In general, genetic correlations were greater in magnitude than respective phenotypic correlations, potentially as GWAS are less sensitive to purely non-genetic factors that may influence phenotypes (e.g. measurement error). As expected we find a highly correlated module of primarily cardiometabolic traits with high density lipoprotein (HDL) showing anti-correlation (*Bulik-Sullivan et al., 2015*). Interestingly, several other traits (gastroesophageal reflux disease (GERD), osteoarthritis) showed moderate genetic correlation to the cardiometabolic trait cluster while atrial fibrillation, for which T2D and CVD are risk factors (*Wasmer et al., 2017*), showed phenotypic, but not genetic, correlation. Our final set of prioritized, age-associated traits included 24 genetically diverse, heritable phenotypes (*Supplementary file 1*). Of these, 11 traits were sufficiently heritable only in UKB, three were sufficiently heritable only among non-UKB meta-analyses, and 10 were well-powered in both UKB and an independent cohort.

## Mitochondrial genes are not enriched among age-related trait GWAS

To test if age-related trait heritability was enriched among mitochondria-relevant loci, we began by simply asking if ~1100 nucDNA genes encoding the mitochondrial proteome from the MitoCarta2.0 inventory (*Calvo et al., 2016*) were found near lead SNPs for our selected traits represented in the NHGRI-EBI GWAS Catalog (https://www.ebi.ac.uk/gwas/) (*MacArthur et al., 2017*) more frequently than expectation (Materials and methods, Appendix 1). To our surprise, no traits showed a statistically significant enrichment of mitochondrial genes (*Figure 2—figure supplement 1A*); in fact, six traits showed a statistically significant depletion. Even more strikingly, MitoCarta genes tended to be nominally enriched in fewer traits than the average randomly selected sample of protein-coding genes (*Figure 2—figure supplement 1B*, empirical p = 0.014). This lack of enrichment was observed more broadly across virtually all traits represented in the GWAS Catalog (*Figure 2—figure supplement 1C*). We also examined specific transcriptional regulators of mitochondrial biogenesis (*TFAM, GABPA, GABPB1, ESRRA, YY1, NRF1, PPARGC1A, PPARGC1B*) and found very little evidence supporting a role for these genes in modifying risk for the age-related GWAS Catalog phenotypes (Appendix 1).

To investigate further, we turned to U.K. Biobank (UKB). We compiled and tested loci encoding the mitochondrial proteome (*Figure 2A*) with which we interrogated the association between common mitochondrial variation and common disease. First, we considered all common variants in or near nucDNA MitoCarta genes, as well as two subsets of MitoCarta: mitochondrial Mendelian disease genes (*Frazier et al., 2019*) and nucDNA-encoded OXPHOS genes. Second, we obtained and tested mtDNA genotypes at up to 213 loci after quality control (Materials and methods) from 360,662 individuals for associations with age-related traits.

First, we used S-LDSC (*Finucane et al., 2015*; *Finucane et al., 2018*) and MAGMA (https://ctg.cncr.nl/software/magma) (*de Leeuw et al., 2015*), two robust methods that can be used to assess gene-based heritability enrichment accounting for LD and several confounders, to test if there was any evidence of heritability enrichment among MitoCarta genes (Materials and methods). We found no evidence of enrichment near nucDNA MitoCarta genes for any trait tested in UKB using S-LDSC (*Figure 2B*, *Figure 2—figure supplement 2A*), consistent with our results from the GWAS Catalog. We replicated this lack of enrichment using MAGMA at two different window sizes (*Figure 2—figure supplement 2C*, *Figure 2—figure supplement 2E*; all *q* > 0.1).

Given the lack of enrichment among the MitoCarta genes, we wanted to (1) verify that our selected methods could detect previously reported enrichments and (2) confirm that common variation in or near MitoCarta genes could lead to expression-level perturbations. We first successfully replicated previously reported enrichment among tissue-specific genes for key traits using both S-LDSC (*Figure 2—figure supplement 3*, *Figure 2—figure supplement 4*) and MAGMA (*Figure 2—figure supplement 5*, *Figure 2—figure supplement 6*, Appendix 1, Materials and methods). We next confirmed that we had sufficient power using both S-LDSC and MAGMA to detect physiologically relevant enrichment effect sizes among MitoCarta genes (*Figure 2—figure supplement 7*, Materials and methods, Appendix 1). We finally examined the landscape of cis-expression QTLs (eQTLs) for these genes and found that almost all MitoCarta genes have cis-eQTLs in at least one tissue and often have cis-eQTLs in more tissues than most protein-coding genes (*Figure 2—figure supplement 8*, Materials and methods, Appendix 1). Hence, our selected methods could detect physiologically relevant heritability enrichments among our selected traits at gene-set sizes

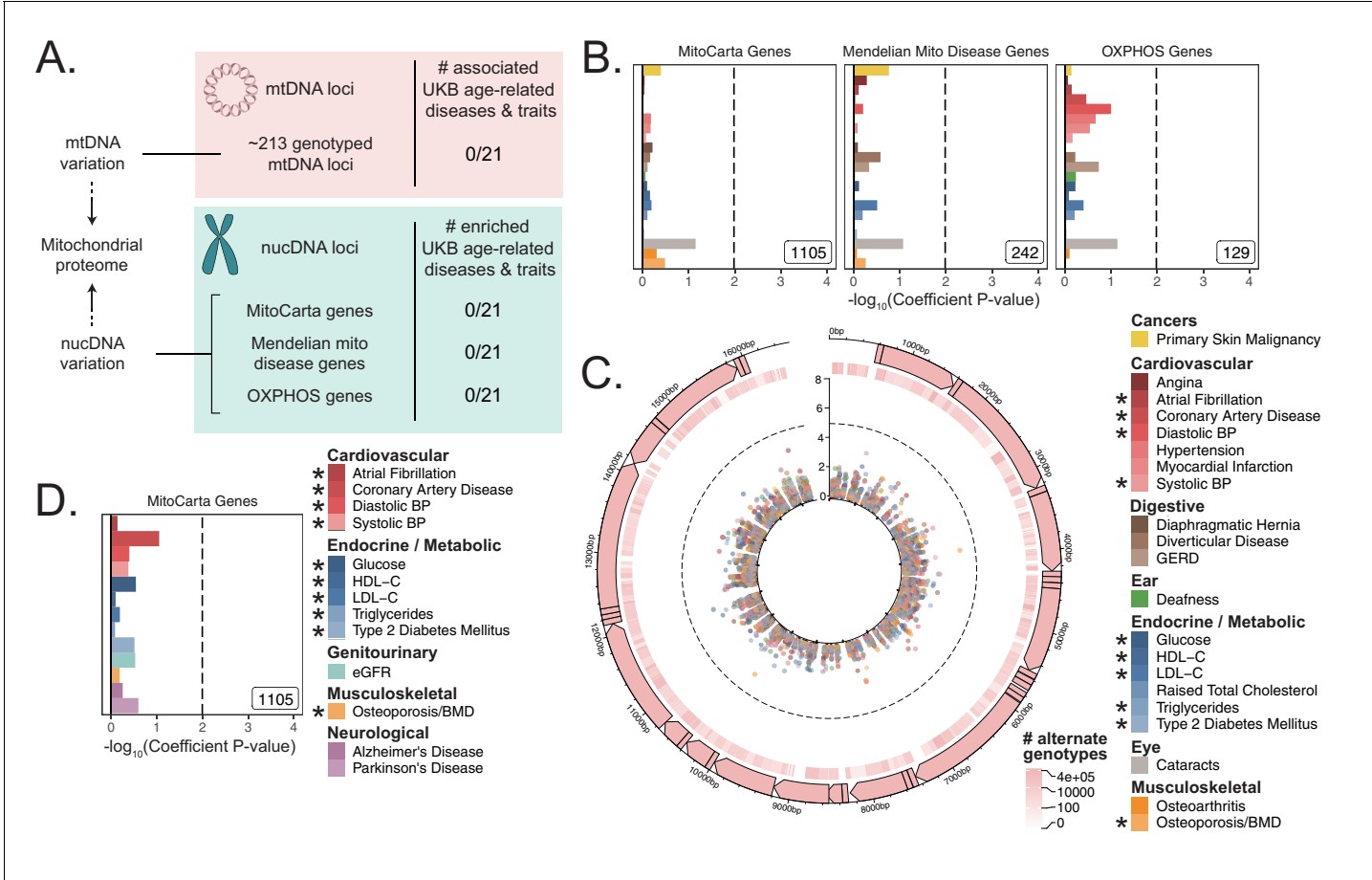

**Figure 2.** Assessment of the association of nucDNA and mtDNA loci contributing to the mitochondrial proteome with age-related traits. (A) Scheme outlining the aspects of mitochondrial function assessed in this study. nucDNA loci contributing to the mitochondrial proteome are shown in teal, while mtDNA loci are shown in pink. (B) S-LDSC enrichment p-values on top of the baseline model in UKB. Inset labels represent gene-set size; dotted line represents BH FDR 0.1 threshold. (C) Visualization of mtDNA variants and associations with age-related diseases. The outer-most track represents the genetic architecture of the circular mtDNA. The heatmap track represents the log-scaled number of individuals with an alternate genotype at each site. The inner track represents mitochondrial genome-wide association p-values, with radial angle corresponding to position on the mtDNA and magnitude representing –$\log_{10}$ p-value. Dotted line represents Bonferroni cutoff for all tested trait-variant pairs. (D) Replication of S-LDSC enrichment results in meta-analyses. Dotted line represents BH FDR 0.1 threshold. * represents traits for which sufficiently well-powered cohorts from both UKB and meta-analyses were available. The trait color legend to the right of panel (C) applies to panels (B) and (C), representing UKB traits. S-LDSC enrichment p-values plotted in (B) and (D) are available in *Source data 1*; mtDNA-GWAS summary statistics are available in *Source data 2*.

The online version of this article includes the following figure supplement(s) for figure 2:

**Figure supplement 1.** Enrichment tests in mitochondria-localizing genes in the GWAS Catalog.

**Figure supplement 2.** Analysis of mitochondria-localizing gene enrichments using alternative methods and cohorts.

**Figure supplement 3.** S-LDSC enrichments for tissues across age-related traits.

**Figure supplement 4.** S-LDSC enrichment coefficients from tissue analysis.

**Figure supplement 5.** MAGMA enrichments for tissues in UKB.

**Figure supplement 6.** MAGMA enrichments for tissues in meta-analyses.

**Figure supplement 7.** Enrichment power of S-LDSC and MAGMA across effect sizes and gene-set sizes.

**Figure supplement 8.** Assessment of cis-eQTLs among mitochondria-localizing genes.

**Figure supplement 9.** Landscape of mtDNA variants and their associations with age-related disease in UKB.

comparable to that of MitoCarta, and common variants in or near MitoCarta genes exerted *cis*-control on gene expression.

Next, we considered mtDNA loci genotyped in UKB, obtaining calls for up to 213 common variants passing quality control across 360,662 individuals (Materials and methods, Appendix 1). We found no significant associations on the mtDNA for any of the 21 age-related traits available in UKB

using linear or logistic regression (Materials and methods, *Figure 2C*, *Figure 2—figure supplement 9*; *Source data 2*).

As a control and to validate our approach, we also performed mtDNA-GWAS for specific traits with previously reported associations. A recent analysis of ~147,437 individuals in BioBank Japan revealed four distinct traits with significant mtDNA associations (*Yamamoto et al., 2020*). Of these, creatinine and aspartate aminotransferase (AST) had sufficiently large sample sizes in UKB. We observed a large number of associations throughout the mtDNA for both traits (p < 1.15 * 10$^{-5}$, *Figure 2—figure supplement 9E*). Thus, our mtDNA association method was able to replicate robust mtDNA associations among well-powered traits.

We sought to replicate our negative results in an independent cohort. We turned to published GWAS meta-analyses (*Teslovich et al., 2010*; *Ehret et al., 2011*; *Manning et al., 2012*; *Morris et al., 2012*; *Schunkert et al., 2011*; *Estrada et al., 2012*; *Christophersen et al., 2017*; *Pattaro et al., 2016*; *Nalls et al., 2019*; *Lambert et al., 2013*; *Supplementary file 1*) and successfully replicated the lack of enrichment for MitoCarta genes across all 10 traits with an available independent cohort GWAS using S-LDSC (*Figure 2D*, *Figure 2—figure supplement 2B*) and MAGMA (*Figure 2—figure supplement 2D*, Appendix 1; all *q* > 0.1). Importantly, while we were unable to pursue analyses for PD and Alzheimer's disease in UKB due to limited case counts, we tested Mito-Carta genes among well-powered meta-analyses for these disorders (Appendix 1) and observed no enrichment (*Figure 2D*; all *q* > 0.1).

In summary, we tested (1) nucDNA loci near genes that encode the mitochondrial proteome in the GWAS Catalog, UKB, and GWAS meta-analyses, (2) transcriptional regulators of mitochondrial biogenesis in the GWAS Catalog, and (3) mtDNA variants in UKB. We found no convincing evidence of heritability enrichment for common age-associated diseases near these mitochondrial loci.

## Of all tested organelles, only the nucleus shows enrichment for age-related trait heritability

We next asked whether heritability for age-related diseases and traits clusters among loci associated with any cellular organelle. We used the COMPARTMENTS database (https://compartments.jensen-lab.org) to define gene-sets corresponding to the proteomes of nine additional organelles (*Binder et al., 2014*) besides mitochondria (Materials and methods). We used S-LDSC to produce heritability estimates for these categories in the UKB age-related disease traits, finding evidence of heritability enrichment in many traits for genes comprising the nuclear proteome (*Figure 3A*, Materials and methods). No other tested organelles showed evidence of heritability enrichment. Variation in or near genes comprising the nuclear proteome explained over 50% of disease heritability on average despite representing only ~35% of tested SNPs (*Figure 3—figure supplement 1*, Appendix 1). We successfully replicated this pattern of heritability enrichment among organelles using MAGMA in UKB at two window sizes (*Figure 3—figure supplement 2A*, *Figure 3—figure supplement 2B*), again finding enrichment only among genes related to the nucleus.

## Much of the nuclear enrichment signal emanates from transcription factors

With over 6000 genes comprising the nuclear proteome, we considered largely disjoint subsets of the organelle's proteome to trace the source of the enrichment signal (*The Gene Ontology Consortium et al., 2019*; *Ashburner et al., 2000*; *Lambert et al., 2018*; *Figure 3B*, Materials and methods, Appendix 1). We found significant heritability enrichment within the set of 1804 genes whose protein products are annotated to localize to the chromosome itself (*q* < 0.1 for nine traits, *Figure 3C*, *Figure 3—figure supplement 3A*). Further partitioning revealed that much of this signal is attributable to the subset classified as TFs (*Lambert et al., 2018*) (1523 genes, *q* < 0.1 for 10 traits, *Figure 3D*, *Figure 3—figure supplement 3B*). We replicated these results using MAGMA in UKB at two window sizes (*Figure 3—figure supplement 2*), and also replicated enrichments among TFs in several (but not all) corresponding meta-analyses (*Figure 3—figure supplement 4*) despite reduced power (*Figure 2—figure supplement 7H*). We generated functional subdivisions of the TFs (Materials and methods, Appendix 1), finding that the non-zinc finger TFs showed enrichment for a highly similar set of traits to those enriched for the whole set of TFs (*Figure 3—figure supplement 5D*, *Figure 3—figure supplement 6B*, *Figure 3—figure supplement 7B*, *Figure 3—figure supplement 8B*).

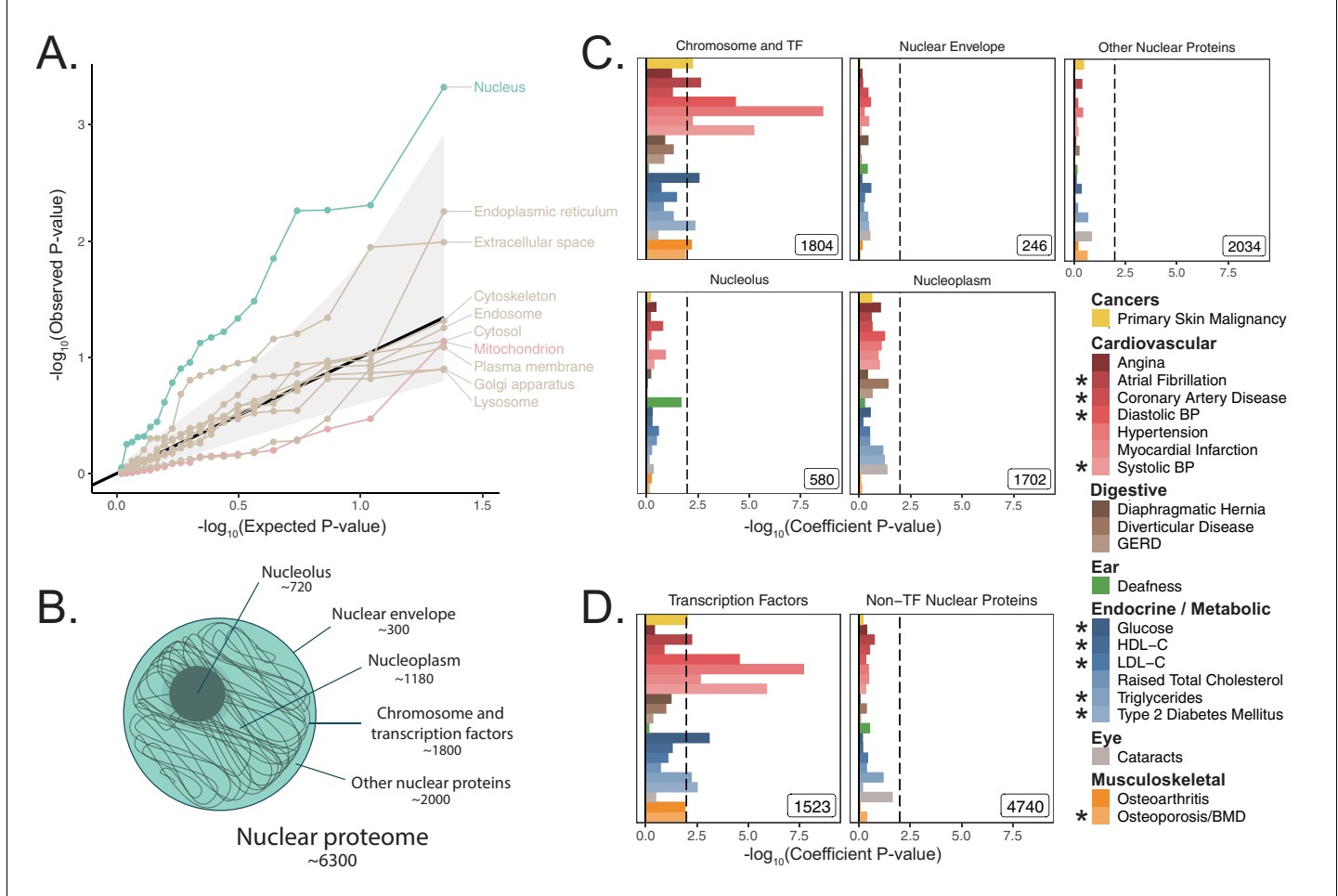

**Figure 3.** Heritability enrichment of organellar proteomes across age-related disease in UK Biobank. (**A**) Quantile-quantile plot of heritability enrichment p-values atop the baseline model for gene-sets representing organellar proteomes, with black line representing expected null p-values following the uniform distribution and shaded ribbon representing 95% CI. (**B**) Scheme of spatially distinct disjoint subsets of the nuclear proteome as a strategy to characterize observed enrichment of the nuclear proteome. Numbers represent gene-set size. (**C**) S-LDSC enrichment p-values for spatial subsets of the nuclear proteome computed atop the baseline model. (**D**) S-LDSC enrichment p-values for TFs and all other nucleus-localizing proteins. Inset numbers represent gene-set sizes, black lines represent cutoff at BH FDR < 10%. * represents traits for which sufficiently well-powered cohorts from both UKB and meta-analyses were available. Enrichment p-values and coefficients are available in *Source data 1*.

The online version of this article includes the following figure supplement(s) for figure 3:

**Figure supplement 1.** Proportions of heritability explained by annotations indicating organellar localization as obtained via S-LDSC.

**Figure supplement 2.** Organelle analysis results in UKB using MAGMA.

**Figure supplement 3.** S-LDSC coefficients from sub-nuclear compartment analysis.

**Figure supplement 4.** Replication of transcription factor enrichment in meta-analyses.

**Figure supplement 5.** MAGMA enrichments of functional subsets of transcription factors in UKB.

**Figure supplement 6.** Replication of enrichment analysis in functional subsets of transcription factors using S-LDSC.

**Figure supplement 7.** S-LDSC enrichment coefficient point estimates in functional subsets of transcription factors corresponding to p-values in *Figure 3—figure supplement 6*.

**Figure supplement 8.** Replication of enrichment analysis in functional subsets of transcription factors using MAGMA with a 100 kb symmetric gene window.

Interestingly, the KRAB domain-containing zinc fingers (KRAB ZFs) (*Kapopoulou et al., 2016*), which are recently evolved (*Figure 3—figure supplement 5H*), were largely devoid of enrichment even compared to non-KRAB ZFs (*Figure 3—figure supplement 5E*, *Figure 3—figure supplement 6C*, *Figure 3—figure supplement 7C*, *Figure 3—figure supplement 8C*). Thus, we find that variation within or near non-KRAB domain-containing TF genes has an outsize influence on age-associated disease heritability.

We next turned to recently published GWAS assessing parental lifespan (*Timmers et al., 2019*) and 'healthspan' via first morbidity hazard (*Zenin et al., 2019*). Both traits showed highly significant heritability via S-LDSC ($h^2(s.e.) = 0.0265$ (0.0019) and 0.0348 (0.003) respectively, Materials and methods). Enrichment analysis of organelles among these traits revealed a significant enrichment for the nucleus for parental lifespan (p = 0.0003) using MAGMA (*Figure 4*). While we observed only a nominally 'suggestive' enrichment for the nucleus for healthspan (p = 0.058), S-LDSC showed significant nuclear heritability enrichment (p = 0.0016, *Figure 4—figure supplement 1*). Analysis of spatial subsets of the nuclear proteome showed significant enrichment for TFs and proteins localizing to the chromosome in both aging phenotypes using MAGMA (*Figure 4*) and for healthspan using S-LDSC (*Figure 4—figure supplement 1*).

## Mitochondrial genes tend to be more 'haplosufficient' than genes encoding other organelles

In light of observing heritability enrichment only among nuclear transcription factors, we wanted to determine if the fitness cost of pLoF variation in genes across cellular organelles mirrored our results. Mitochondria-localizing genes and TFs play a central role in numerous Mendelian diseases (*Frazier et al., 2019*; *Jimenez-Sanchez et al., 2001*; *Worman and Courvalin, 2002*; *Cleaver, 1994*), so we initially hypothesized that genes belonging to either category would be under significant purifying selection (i.e., constraint). We obtained constraint metrics from gnomAD (https://gnomad.broadinstitute.org) (*Karczewski et al., 2020*) as the LoF observed/expected fraction (LOEUF). In agreement with our GWAS enrichment results, we observed that the mitochondrion on average is one of the least constrained organelles we tested, in stark contrast to the nucleus (*Figure 5A*). In fact, the nucleus was second only to the set of 'haploinsufficient' genes (defined based on curated human clinical genetic data; *Karczewski et al., 2020*, Materials and methods) in the proportion of its genes in the most constrained decile, while the mitochondrion lay on the opposite end of the spectrum (*Figure 5B*). Interestingly, even the Mendelian mitochondrial disease genes had a high tolerance to pLoF variation on average in comparison to TFs (*Figure 5C*). Even across different categories of TFs, we observed that highly constrained TF subsets tend to show GWAS enrichment (*Figure 5-Figure supplement 1*, *Figure 3-Figure supplement 5E*) relative to unconstrained subsets

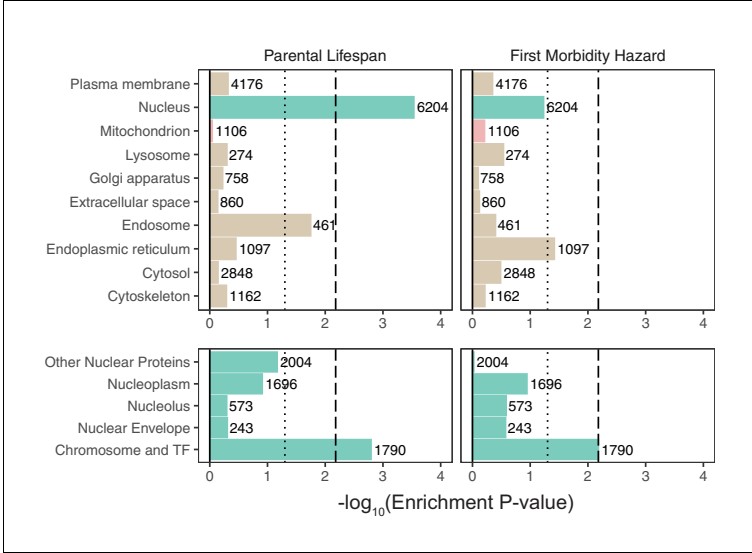

**Figure 4.** Enrichment of organellar proteomes within parental lifespan and healthspan as proxies for aging. Upper panels represent organelle proteomes; lower panels represent spatial subsets of the nuclear proteome. Numbers atop each bar represent gene-set sizes. Dashed lines represent cutoff at BH FDR < 10%, dotted lines represent nominal p = 0.05. p-Values and coefficients available in *Source data 3*.

The online version of this article includes the following figure supplement(s) for figure 4:

**Figure supplement 1.** Enrichment of organellar proteomes within parental lifespan and healthspan as proxies for aging using S-LDSC.

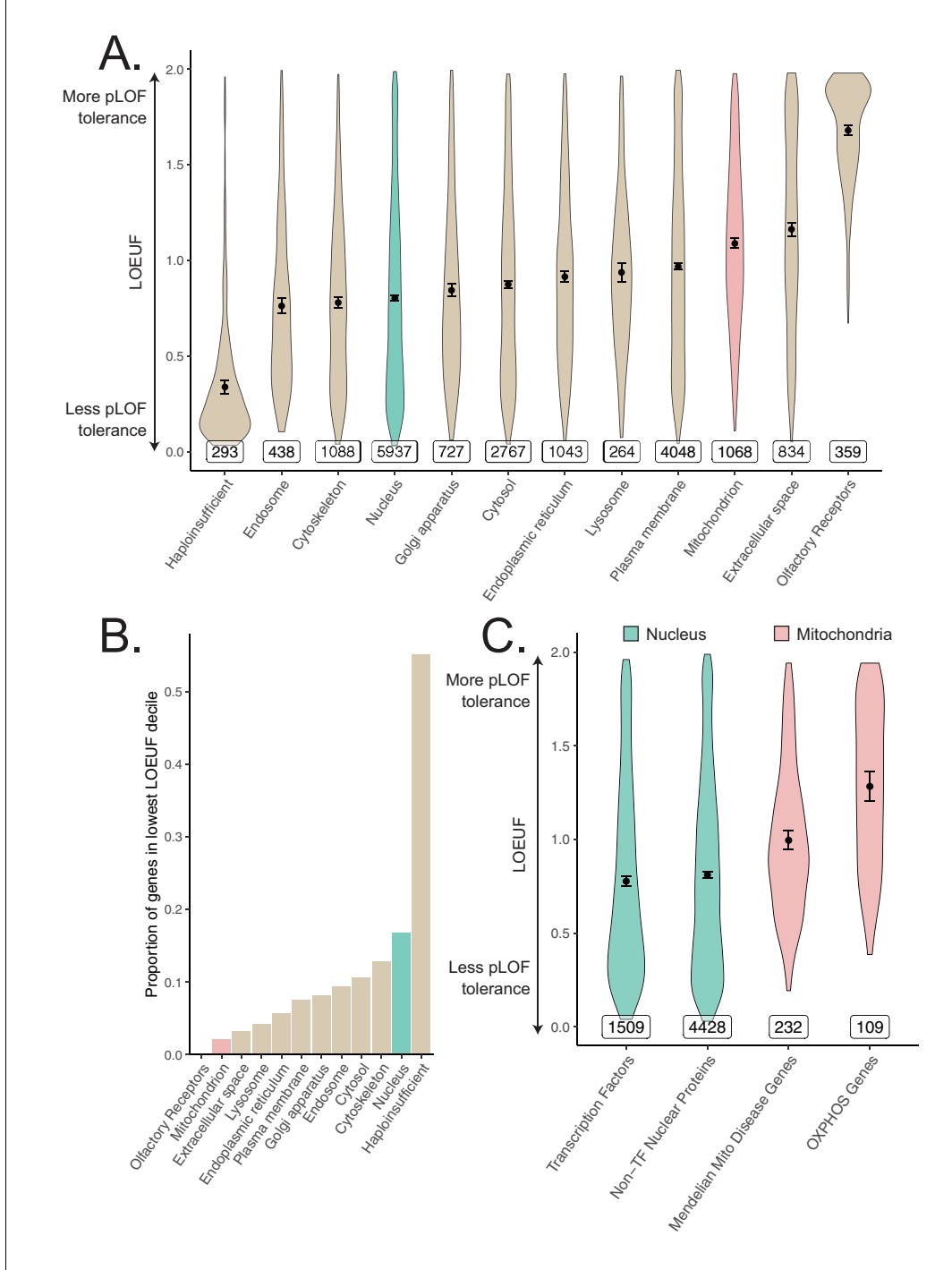

**Figure 5.** Differences in constraint distribution across organelles. (**A**) Constraint as measured by LOEUF from gnomAD v2.1.1 for genes comprising organellar proteomes, book-ended by distributions for known haploinsufficient genes as well as olfactory receptors. Lower values indicate genes exacting a greater organismal fitness cost from a heterozygous LoF variant (greater constraint). (**B**) Proportion of each gene-set found in the lowest LOEUF decile. Higher values indicate gene-sets containing more highly constrained genes. (**C**) Constraint distributions for subsets of the nuclear-encoded mitochondrial proteome (red) and subsets of the nucleus (teal). Black points represent the mean with 95% CI. Inset numbers represent gene-set size.

The online version of this article includes the following figure supplement(s) for figure 5:

**Figure supplement 1.** Constraint distributions within subdivisions of the nuclear proteome.

*Figure 5 continued on next page*

*Figure 5 continued*

**Figure supplement 2.** Enrichment results across age-related disease in UKB after correcting for constraint using MAGMA with a 5 kb up, 1.5 kb down window.

**Figure supplement 3.** Enrichment results across age-related disease in UKB after correcting for gene constraint using MAGMA with a 100 kb symmetric gene window.

for our tested traits. Indeed, explicit inclusion of LOEUF as a covariate in the enrichment analysis model (Materials and methods) reduced the significance of (but did not eliminate) the enrichment seen for the TFs (*Figure 5-Figure supplement 2B*, *Figure 5-Figure supplement 2E*, *Figure 5-Figure supplement 2F*). Thus, while disruption in both mitochondrial genes and TFs can produce rare disease, the fitness cost of heterozygous variation in mitochondrial genes appears to be far lower than that among TFs. This dichotomy reflects the contrasting enrichment results between mitochondrial genes and TFs and supports the importance of gene regulation as it relates to evolutionary conservation.

## Discussion

Pathology in cellular organelles has been widely documented in age-related diseases (*López-Otín et al., 2013*; *Ozcan et al., 2004*; *Colacurcio and Nixon, 2016*; *Kanfi et al., 2010*; *Blasco, 2007*; *Bhattarai et al., 2020*). Using a human genetics approach, here we report the unexpected discovery that except for the nucleus, cellular organelles tend not to be enriched in genetic associations for common, age-related diseases. We started with a focus on the mitochondria as a decline in mitochondrial abundance and activity has long been reported as one of the most consistent correlates of aging (*Wanagat et al., 2001*; *Fleischman et al., 2010*; *Trounce et al., 1989*; *Taylor et al., 2003*) and age-associated diseases (*Petersen et al., 2004*; *Mootha et al., 2003*; *Schapira et al., 1990*; *Bender et al., 2006*; *Ashar et al., 2017*; *Fannin et al., 1999*; *Kelley et al., 2002*; *Patti et al., 2003*; *Stump et al., 2003*). We tested common variants contributing to the mitochondrial proteome on the nucDNA and mtDNA and found no convincing evidence of heritability enrichment in any tested trait, cohort, or method. We systematically expanded our analysis to survey 10 organelles and found that only the nucleus showed enrichment, with much of this signal originating from nuclear TFs. Constraint analysis showed a substantial fitness cost to heterozygous loss-of-function mutations in genes encoding the nuclear proteome, whereas genes encoding the mitochondrial proteome were 'haplosufficient'.

Here, we focus on enrichment to place the complex genetic architectures of age-related traits in a broader biological context and prioritize pathways for follow-up. For these highly polygenic traits, any large fraction of the genome may explain a statistically significant amount of disease heritability (*de Leeuw et al., 2016*; *Loh et al., 2015*), and indeed associations between individual organelle-relevant loci and certain common diseases have been identified previously (*Billingsley et al., 2019*; *Kraja et al., 2019*). For example, variants in the endoplasmic reticular genes *WFS1* and *ATF6B* and the mitochondrial gene *ATP5G1* have been associated with common T2D (*Xue et al., 2018*). These genes are present in the respective organelle gene-sets, however unlike TFs, neither the endoplasmic reticulum nor the mitochondrion showed enrichment for T2D. Importantly, both MAGMA and S-LDSC are capable of detecting an enrichment even in a highly polygenic background. Both methods have been used in the past to identify biologically plausible disease-relevant tissues (*Finucane et al., 2015*; *Finucane et al., 2018*) and pathway enrichments (*Jansen et al., 2019*; *Pardiñas et al., 2018*) in traits across the spectrum of polygenicity, and we identify enrichments among disease-relevant tissues using both methods in several highly polygenic traits.

While previous work has shown that common disease GWAS can be enriched for expression in specific disease-relevant organs (*Finucane et al., 2018*; *Maurano et al., 2012*), our data suggest that this framework does not generally extend from organs to organelles. This finding contrasts with our classical nosology of inborn errors of metabolism that tend to be mapped to 'causal' organelles, for example, lysosomal storage diseases, disorders of peroxisomal biogenesis, and mitochondrial OXPHOS disorders. The observed enrichment for TFs within the nucleus indicates that common variation influencing genome regulation impacts common disease risk more than variation influencing individual organelles.

Our analysis of common inherited mitochondrial variation represents, to our knowledge, the most comprehensive joint assessment of mitochondria-relevant nucDNA and mtDNA variation in age-related diseases. We replicated mtDNA associations with creatinine and AST observed previously in BioBank Japan (*Yamamoto et al., 2020*), further supporting our approach. While individual mtDNA variants have been previously associated with certain traits (*Raule et al., 2007*; *Yu et al., 2008*; *Hudson et al., 2013a*), these associations appear to be conflicting in the literature, perhaps because of limited power and/or uncontrolled confounding biases such as population stratification (*Samuels et al., 2006*; *Biffi et al., 2010*). Our negative results are surprising, but they are compatible with a prior enrichment analysis focused on T2D (*Segrè et al., 2010*) as well as a small number of isolated reports interrogating either mitochondria-relevant nucDNA (*Segrè et al., 2010*) or mtDNA (*Yamamoto et al., 2020*; *Saxena et al., 2006*; *Hudson et al., 2014*; *Hudson et al., 2013b*) loci in select diseases.

To our knowledge, we are the first to systematically document heterogeneity in average pLoF across cellular organelles. That MitoCarta genes are 'haplosufficient' and pLoF tolerant (*Figure 5A*) is consistent with the observation that most of the ~300 inborn mitochondrial disease genes produce disease with recessive inheritance (*Frazier et al., 2019*) and healthy parents. The few mitochondrial disorders that show autosomal dominant inheritance are nearly always due to dominant negativity rather than haploinsufficiency. The intolerance of TFs to pLoF variation (*Figure 5C*) provide a stark contrast to the results from the mitochondria that is borne out in their associated Mendelian disease syndromes: TFs are known to be haploinsufficient (*Seidman and Seidman, 2002*) and even regulatory variants modulating their expression can produce severe Mendelian disease (*van der Lee et al., 2020*). We observe enrichment among TFs for 10 different diseases as well as parental lifespan and healthspan, consistent with observed elevated purifying selection against pLoF variants in these genes. Our enrichment results combined with pLoF intolerance suggest that variation among TFs may produce disease-associated variants with larger effect sizes than expectation, underscoring their importance as genetic 'levers' for common disease heritability.

Why are mitochondria so robust to variation in gene dosage (*Figure 5*) and hence 'haplosufficient?' We propose two possibilities. First, mitochondrial pathways tend to be highly interconnected, and it was already proposed by *Wright, 1934* and later by *Kacser and Burns, 1981* that haplosufficiency arises as a consequence of physiology, that is, system output is inherently buffered against the partial loss of a single gene due to the network organization of metabolic reactions. Kacser and Burns in fact explicitly mention that noncatalytic gene products fall outside their framework, and we believe that our finding that nucleus-localizing and cytoskeletal genes are the two most pLoF-intolerant compartments is consistent with their assessment. Second, mitochondria were formerly autonomous microbes and hence may have retained vestigial layers of 'intra-organelle buffering' against genetic variation. Numerous feedback control mechanisms, including respiratory control (*Chance and Williams, 1955*), help to ensure organelle robustness across physiological extremes (*Vafai and Mootha, 2012*; *Balaban et al., 1986*). In fact, a recent CRISPR screen showed that of the genes for which knock-out modified survival under a mitochondrial poison, there is a striking over-representation of genes that themselves encode mitochondrial proteins (*To et al., 2019*).

Throughout this study, we have tested for enrichment among inherited common variant associations near genes via an additive genetic model. We acknowledge the limitations of focusing on a specific genetic model and variant frequency regime, though note that common variation is the largest documented source of narrow-sense heritability, which typically accounts for a majority of disease heritability (*Golan et al., 2014*; *Polderman et al., 2015*). First, we consider only common variants. While rare variants may prove to be instructive, it is notable that a previous rare variant analysis in T2D (*Fuchsberger et al., 2016*) failed to show enrichment among OXPHOS genes. Second, we consider only additive genetic models. A recessive model may be particularly fruitful for mitochondrial genes given their tolerance to pLoF variation, however these models are frequently power-limited and may not explain much more phenotypic variance than additive models (*Hill et al., 2008*; *Zhu et al., 2015*). Third, we have not considered epistasis. The effects of mtDNA-nucDNA interactions (*Rand and Mossman, 2020*) in common diseases have yet to be assessed. While there is debate about whether biologically-relevant epistasis can be simply captured by main effects (*Polderman et al., 2015*; *Hill et al., 2008*; *Sackton and Hartl, 2016*; *Hemani et al., 2014*) at individual loci, it is possible that modeling mtDNA-nucDNA interactions will reveal new contributions. Fourth, to systematically assess all organelles, we restrict our analyses to variants near genes

comprising each organelle's proteome. It remains possible that future work will systematically identify novel organelle-relevant loci elsewhere in the genome which contribute disproportionately to age-related trait heritability. Fifth, while we are well-powered to detect physiologically relevant enrichments among most tested organelles (including the mitochondrion), our power may be more limited for particularly small compartments (e.g. lysosome). Finally, it is crucial not to confuse our mtDNA-GWAS results with previously reported associations between somatic mtDNA mutations and age-associated disease (*Bender et al., 2006*; *Wanagat et al., 2001*; *Taylor et al., 2003*) – the present work is focused on germline variation.

We have not formally addressed the causality of mitochondrial dysfunction in common age-related disease and the observed lack of heritability enrichment does not preclude the possibility of a therapeutic benefit in targeting the mitochondrion for age-related disease. For example, mitochondrial dysfunction is documented in brain or heart infarcts following blood vessel occlusion in laboratory-based models (*Solenski et al., 2002*; *Flameng et al., 1991*). Clearly, mitochondrial genetic variants do not influence infarct risk in this laboratory model, but pharmacological blockade of the mitochondrial permeability transition pore can mitigate reperfusion injury and infarct size (*Weinbrenner et al., 1998*). Future studies will be required to determine if and how the mitochondrial dysfunction associated with common age-associated diseases can be targeted for therapeutic benefit. Efforts to develop reliable measures of mitochondrial function and dysfunction have the potential to unbiasedly discover genetic instruments that influence the mitochondrion, and causal inference techniques such as Mendelian Randomization may shed light on this important causal question.

Our finding that the nucleus is the only organelle that shows enrichment for common age-associated trait heritability builds on prior work implicating nuclear processes in aging. Most human progeroid syndromes result from monogenic defects in nuclear components (*Kubben and Misteli, 2017*) (e.g. *LMNA* in Hutchinson-Gilford progeria syndrome, *TERC* in dyskeratosis congenita), and telomere length has long been observed as a marker of aging (*Garcia et al., 2007*). Heritability enrichment of age-related traits among gene regulators is consistent with the epigenetic dysregulation (*Han and Brunet, 2012*) and elevated transcriptional noise (*López-Otín et al., 2013*; *Bahar et al., 2006*) observed in aging (e.g. *SIRT6* modulation influences mouse longevity and metabolic syndrome; *Kanfi et al., 2012*; *Kanfi et al., 2010*). An important role for gene regulation in common age-related disease is in agreement with both the observation that a very large fraction of common disease-associated loci corresponds to the non-coding genome and the enrichment of disease heritability in histone marks and TF binding sites (*Finucane et al., 2015*; *Karczewski et al., 2013*). Given that a deterioration in several other cellular organelles has been so frequently documented in age-related traits, a future challenge lies in elucidating how inherited variation in or near TFs ultimately leads to the observed organelle dysfunction in age-related disease.

## Data availability

Heritability point estimates and standard errors for age-related traits are listed in *Supplementary file 1*. Genetic and phenotypic correlation point estimates and standard errors/p-values plotted in *Figure 1B* are available in *Figure 1—source data 1*. Summary statistics from mtDNA-GWAS (plotted in *Figure 2* and *Figure 2—figure supplement 9*) are available in *Source data 2*. All gene-based enrichment analysis p-values and point estimates are available in *Source data 1* and *Source data 3*. Period prevalence data for diseases in the UK can be obtained from *Kuan et al., 2019*. Gene-sets can be found using COMPARTMENTS (https://compartments.jensenlab.org), MitoCarta 2.0 (https://www.broadinstitute.org/files/shared/metabolism/mitocarta/human.mitocarta2.0.html), *Lambert et al., 2018* (DOI: 10.1016/j.cell.2018.01.029), *Frazier et al., 2019* (DOI: 10.1074/jbc.R117.809194), *Finucane et al., 2018* (https://alkesgroup.broadinstitute.org/LDSCORE/), *Kapopoulou et al., 2016* (DOI: 10.1111/evo.12819), and the MacArthur laboratory (https://github.com/macarthur-lab/gene_lists, copy archived at swh:1:rev:fcc849637bd71e683bff-c618e1a48081a8df08f8), *Minikel, 2021*. Gene age estimates were obtained from *Litman and Stein, 2019* (DOI: 10.1053/j.seminoncol.2018.11.002). GWAS catalog annotations can be obtained from: https://www.ebi.ac.uk/gwas. Heritability estimates across UKB can be obtained at: https://nealelab.github.io/UKBB_ldsc/. UKB summary statistics can be obtained from Neale lab GWAS round 2: https://github.com/Nealelab/UK_Biobank_GWAS, (copy archived at swh:1:rev:dc7b7b590413e-c96a45a64f7213f50a3a0606198), *Howrigan, 2021*. Annotations for the Baseline v1.1 and BaselineLD

v2.2 models as well as other relevant reference data, including the 1000G EUR reference panel, can be obtained from https://alkesgroup.broadinstitute.org/LDSCORE/. eQTL and expression data in human tissues can be obtained from GTEx: https://www.gtexportal.org. Constraint estimates can be found via gnomAD: https://gnomad.broadinstitute.org. See citations for publicly available GWAS meta-analysis summary statistics (*Teslovich et al., 2010*; *Ehret et al., 2011*; *Timmers et al., 2019*; *Zenin et al., 2019*; *Manning et al., 2012*; *Morris et al., 2012*; *Schunkert et al., 2011*; *Estrada et al., 2012*; *Christophersen et al., 2017*; *Pattaro et al., 2016*; *Nalls et al., 2019*; *Lambert et al., 2013*).

## Code availability

Our analysis leverages publicly available tools including LDSC for heritability enrichment and genetic correlation (https://github.com/bulik/ldsc, copy archived at swh:1:rev:aa33296abac9569a6422ee6-ba7eb4b902422cc74); *Schorsch, 2021*, MAGMA v1.07b for gene-set enrichment analysis (https://ctg.cncr.nl/software/magma), Hail v0.2.51 for distributed computing and mtDNA GWAS (https://hail.is), the R circlize package (*Gu et al., 2014*) for visualization of mtDNA-GWAS, and the R polycor package for phenotypic correlations with binary traits.

# Materials and methods

## Trait selection

Sex-standardized period prevalence of over 300 diseases was obtained from an extensive survey of the National Health Service in the UK as reported previously (*Kuan et al., 2019*). To select high prevalence late-onset diseases, we ranked diseases with a median onset over 50 years of age by the sum of the period prevalence of all age categories above 50. We selected the top 30 diseases using this metric and manually mapped these traits to similar or equivalent phenotypes with publicly available summary statistics from UKB and/or well-powered meta-analyses (e.g. Parkinson's Disease and Alzheimer's Disease for dementia) resulting in 24 traits with data available in UKB (RRID:SCR_012815), meta-analyses, or both (*Supplementary file 1*).

## Criteria for inclusion of summary statistics

We manually mapped selected age-related diseases and traits to corresponding phenotypes in UKB. In parallel, we searched the literature to identify well-powered EUR-predominant GWAS (referred to as meta-analyses) that (1) used primarily non-targeted arrays, (2) had publicly available full summary statistics, and (3) did not enroll individuals from UKB to serve as independent replication (Appendix 1). We produced heritability estimates using stratified linkage-disequilibrium score regression (S-LDSC, https://github.com/bulik/ldsc) (*Finucane et al., 2015*) atop the BaselineLD v2.2 model using reference LD scores computed from 1000G EUR (https://alkesgroup.broadinstitute.org/LDSCORE/). We computed the heritability Z-score, a statistic that captures sample size, polygenicity, and heritability (*Finucane et al., 2015*), and included only traits with heritability Z-score > 4 (Appendix 1) for further analysis.

## Genetic correlations among age-related traits

Pairwise genetic correlations, $r_g$, were computed using linkage-disequilibrium score correlation (*Bulik-Sullivan et al., 2015*) on all selected age-related traits with heritability Z-score > 4. We used UKB summary statistics (https://github.com/Nealelab/UK_Biobank_GWAS) for all sufficiently powered traits; summary statistics from meta-analyses were used for eGFR (*Pattaro et al., 2016*), Alzheimer's Disease (*Lambert et al., 2013*), and Parkinson's Disease (*Nalls et al., 2019*) as these traits showed heritability Z-score > 4 within meta-analyses but not in UKB (*Supplementary file 1*). p-Values for genetic correlation represented deviation from the null hypothesis $r_g = 0$. Traits were ordered by their contribution to the first eigenvector of the absolute value of the correlation matrix, with point estimates and standard errors available in *Source data 1*. Bonferroni correction was applied producing a p-value cutoff of $0.05/[242 + 212] = 1.03 * 10^{-4}$, accounting for both genotypic and phenotypic correlation hypothesis tests.

## Phenotypic correlations in UKB

Pairwise phenotypic correlations, $r_p$, were computed for all 21 traits with well-powered individual level data available in UKB (*Supplementary file 1*). Pearson correlation was computed between continuous traits via cor.test in R (RRID:SCR_001905) with a two-sided alternative. Tetrachoric correlation was used to compute correlations between binary traits and biserial correlation was used for correlations between binary and continuous traits, using the polychor and polyserial functions of the polycor package in R using the two-step approximation, respectively. These approaches model a latent normally distributed variable underlying binary traits. p-Values were computed using a normal approximation using standard error estimates from polycor. Point estimates and standard errors are available in *Figure 1—source data 1*.

## Assessment of mitochondria-localizing genes in the GWAS catalog

We mapped variants in the GWAS Catalog (RRID:SCR_012745) (obtained on September 5th, 2019, https://www.ebi.ac.uk/gwas/) meeting genome-wide significance (p < 5e-8) to genes using provided annotations, producing a set of trait-associated genes for each trait. We manually selected phenotypes represented in the GWAS Catalog matching our set of age-associated traits with > 30 trait-associated genes. For each trait, we computed the proportion of trait-associated genes that were mitochondria-localizing (defined via MitoCarta2.0; *Calvo et al., 2016*, RRID:SCR_018165) and tested for enrichment or depletion relative to overall genome background using two-sided Fisher's exact tests. We corrected for multiple hypothesis tests with the Benjamini-Hochberg (BH) procedure at FDR q-value < 0.1.

We also computed the test statistic $N_g^{enrich}$, defined as the number of age-associated traits showing a nominal (not necessarily statistically significant) enrichment for a given gene-set $g$, for the MitoCarta genes. We then generated an empirical null distribution for $N_g^{enrich}$. We drew 1000 random samples of protein-coding genes, where each sample contained the same number of genes as the set of mitochondria-localizing genes and computed $N_g^{enrich}$ for each of these gene-sets (*Figure 2—figure supplement 1B*). The one-sided p-value, defined as $\Pr\left(N_g^{enrich} \leq x\right)$ under the null, was subsequently obtained.

We expanded our enrichment/depletion analysis to all 332 traits in the GWAS Catalog with over 30 trait-associated genes; for enrichment or depletion testing, we used two-sided Fisher's exact tests and corrected for multiple hypothesis testing with the BH procedure at FDR q-value < 0.1.

## Harmonization and filtering of summary statistics for LDSC and MAGMA

UKB summary statistics previously formatted for use with LDSC and filtered to HapMap3 (HM3) (RRID:SCR_004563) SNPs (https://github.com/Nealelab/UKBB_ldsc) were used for analysis with S-LDSC. For analysis with MAGMA v1.07b (*de Leeuw et al., 2015*), we included variants from the full Neale Lab UKB Round 2 GWAS summary statistics (https://github.com/Nealelab/UK_Biobank_GWAS) with INFO > 0.8 and MAF > 0.01, and excluded any variants flagged as low confidence (a heuristic defined by MAF < 0.001 or expected case MAC < 25).

Summary statistics obtained from publicly available GWAS meta-analyses (*Teslovich et al., 2010*; *Ehret et al., 2011*; *Manning et al., 2012*; *Morris et al., 2012*; *Schunkert et al., 2011*; *Estrada et al., 2012*; *Christophersen et al., 2017*; *Pattaro et al., 2016*; *Nalls et al., 2019*; *Lambert et al., 2013*) were reported in varied formats. We manually verified the genome build upon which each meta-analysis reported results and ensured that all sets of summary statistics contained columns listing p-value, variant rsID, genome-build specific coordinates, and if available, variant-specific sample size (*Supplementary file 1*). If variant coordinates or rsID were not provided, the relevant columns were obtained from dbSNP (RRID:SCR_002338) database version 130 (for hg18) or 146 (for hg19). We used the summary statistic munging script provided with S-LDSC (https://github.com/bulik/ldsc) to generate summary statistics compatible with S-LDSC, restricting to HM3 SNPs as these tend to be best behaved for analysis with LDSC. For use of meta-analyses with MAGMA (*de Leeuw et al., 2015*), we restricted analysis to variants with INFO > 0.8 and MAF > 0.01 if such information was provided.

## Multiple testing correction for gene-set enrichment analysis

To account for the multiple hypothesis tests performed throughout this study for age-related traits, we obtained p-value thresholds via the BH procedure at FDR < 0.1 for all gene-sets assessed for a given method and cohort type (where the two cohort types were UKB and meta-analysis). The BH procedure at FDR < 0.1 was also applied to our analyses of parental lifespan and healthspan.

## Gene-set-based enrichment analysis

We extensively use S-LDSC and MAGMA to perform gene-set enrichment analyses among GWAS summary statistics. To test enrichment with S-LDSC, SNPs were mapped to each gene with a 100 kb symmetric window as recommended (*Finucane et al., 2018*) and LD scores were computed using the 1000G EUR reference panel (RRID:SCR_006828) (https://alkesgroup.broadinstitute.org/LDSCORE/) and subsequently restricted to the HM3 SNPs. We used S-LDSC to test for heritability enrichment controlling for 53 annotations including coding regions, enhancer regions, 5' and 3' UTRs, and others as previously described (*Finucane et al., 2015*) (baseline v1.1, referred to as baseline model hereafter). We also used MAGMA with both 5 kb up, 1.5 kb down and 100 kb symmetric windows to test for enrichment. MAGMA gene-level analysis was performed with the 1000G EUR LD reference panel to account for LD structure, and gene-set analysis was performed including covariates for gene length, variant density, inverse minor allele count (MAC), as well as log-transformed versions of these covariates. Statistical tests for both S-LDSC and MAGMA were one-sided, considering enrichment only. For both methods, we included the relevant superset of genes as a control to ensure that our analysis was competitive (Appendix 1). We refer to this approach as the 'usual approach.' All enrichment effect size estimates and p-values are available in *Source data 1* and *Source data 3*.

## Enrichment analysis of genes comprising the mitochondrial proteome

We obtained the set of nuclear-encoded mitochondria-localizing genes using MitoCarta2.0 (*Calvo et al., 2016*) and used the literature to obtain the subset of MitoCarta genes involved in inherited mitochondrial disease (*Frazier et al., 2019*) as well as those producing components of oxidative phosphorylation (OXPHOS) complexes. We used both S-LDSC and MAGMA to test for enrichment in the usual way (Materials and methods) controlling for the set of protein-coding genes to ensure a competitive analysis (Appendix 1). We also tested mitochondria-localizing genes for enrichment in meta-analyses using S-LDSC and MAGMA with the same parameters as for UKB traits (Appendix 1).

## Tissue-expressed gene-set enrichment analysis

To obtain the set of genes most expressed in a given tissue versus others, we obtained t-statistics computed from GTEx (RRID:SCR_013042) v6 gene-level transcript-per-million (TPM) data corrected for age and sex as published previously (*Finucane et al., 2018*). For each tissue, we selected the top 2485 genes (10%) with the highest t-statistics for tissue-specific expression, producing tissue-expressed gene-sets. We selected nine tissues based on expectation of enrichment for our tested traits in UKB (e.g. liver for LDL levels, esophageal mucosa for GERD). We used both S-LDSC and MAGMA to test for enrichment in the usual way (Materials and methods) controlling for the set of tissue-expressed genes to ensure a competitive analysis (Appendix 1). Tissue-expressed gene-set analyses were performed on meta-analyses with S-LDSC and MAGMA on the same tissues using the same parameters as used in UKB.

## Power analysis

To test for the effects of gene-set size on power, we selected 10 positive control tissue-trait pairs based on (1) the presence of tissue enrichment in UKB with S-LDSC and MAGMA and (2) if the observed enrichment was biologically plausible. The pairs tested were liver-HDL, liver-LDL, liver-TG, liver-cholesterol, pancreas-glucose, pancreas-T2D, atrial appendage-atrial fibrillation, sigmoid colon-diverticular disease, coronary artery-myocardial infarction, and visceral adipose-HDL. We then, in brief, used an empirical sampling-based approach, generating random subsamples of a selected set of tissue-expressed gene-sets at four different gene-set sizes (1523, 1105, 800, and 350 genes), defining power as the proportion of trials showing a significant enrichment (Appendix 1). We used

the same sub-sampled gene-sets for enrichment analysis using both S-LDSC and MAGMA in the usual way (Materials and methods) controlling for the set of tissue-expressed genes to ensure a competitive analysis (Appendix 1). We used the same gene-sets among the subset of the positive control traits that showed enrichment in the corresponding meta-analysis to verify power for the meta-analyses (Appendix 1).

## Cross-tissue eQTL analysis

We obtained the set of eGenes from GTEx (RRID:SCR_013042) v8 across 49 tissues (https://www.gtexportal.org), filtering to only include cis-eQTLs with q-value < 0.05. To determine how the landscape of cis-eQTLs for MitoCarta genes compared to other protein-coding genes, we regressed the number of tissues with a detected cis-eQTL for a given gene x, $N_x^{eQTL}$, onto an indicator for membership in a given organellar proteome ($I_x^{organelle}$), controlling for gene length, log gene length, breadth of expression ($\tau_x$), and the number of tissues with detected expression > 5 TPM ($N_x^{express}$, Appendix 1). To quantify breadth of expression, we obtained median-per-tissue GTEx v8 TPM expression values and computed $\tau$ (*Yanai et al., 2005*) after removing lowly expressed genes with maximal cross-tissue TPM < 1, defined as:

$$\tau_x = \frac{\sum_{i=1}^{n}(1 - \hat{x}_i)}{n - 1} \text{ where } \hat{x}_i = \frac{x_i}{\max\limits_{1 \le i \le n} x_i}$$

where $x_i$ is the expression of gene $x$ in tissue $i$ with $n$ tissues. $\tau$ ranges from 0 to 1, with lower $\tau$ indicating broadly expressed genes and higher $\tau$ indicating more tissue specific expression patterns. Because GTEx sampled multiple tissue subtypes (e.g. brain sub-regions) that show correlated expression profiles (*Melé et al., 2015*) which bias $\tau_x$, $N_x^{eQTL}$, and $N_x^{express}$ upward, for each broader tissue class (brain, heart, artery, esophagus, skin, cervix, colon, adipose), we selected a single representative tissue when computing these quantities (*Figure 3—figure supplement 5B*, Appendix 1). We used LD scores computed from the 1000G EUR reference panel. The model, fit via ordinary least squares for each tested organelle, was:

$$N_x^{eQTL} \sim I_x^{organelle} + N_x^{express} + \tau_x + \log(genelength) + genelength$$

## mtDNA-wide association study

We obtained mtDNA genotype data on 265 variants as obtained on the UK Biobank Axiom array and the UK BiLEVE array from the full UKB release (RRID:SCR_012815) (*Sudlow et al., 2015*). To perform variant QC, we used evoker-lite (RRID:SCR_009145) (*Morris et al., 2010*) to generate fluorescence cluster plots per-variant and per-batch and manually inspected the results, removing 19 variants due to cluster plot abnormalities (*Supplementary file 2a*, Appendix 1). We additionally removed any variants with heterozygous calls, within-array-type call rate < 0.95, and with less than 20 individuals with an alternate genotype. For case-control traits, we removed any phenotype-variant pair with an expected case count of alternate genotype individuals of less than 20, resulting in a maximum of 213 variants tested per trait (Appendix 1). To perform sample QC, we restricted samples to the same samples from which UKB summary statistics were generated (https://github.com/Nealelab/UK_Biobank_GWAS), namely unrelated individuals seven standard deviations away from the first 6 European sample selection PCs with self-reported white-British, Irish, or White ethnicity and no evidence of sex chromosome aneuploidy. We additionally removed any samples with within-array-type mitochondrial variant call rate < 0.95, resulting in 360,662 unrelated samples of EUR ancestry. We generated the LD matrix for mitochondrial DNA variants using Hail v0.2.51 (https://hail.is) pairwise for all 213 variants tested across all post-QC samples.

We ran mtDNA-GWAS for all 21 UKB age-related phenotypes as well as creatinine and AST using Hail v0.2.51 via linear regression controlling for the first 20 PCs of the nuclear genotype matrix, sex, age, age$^2$, sex*age, and sex*age$^2$ as performed for the UKB GWAS (https://github.com/Nealelab/UK_Biobank_GWAS). We also used Hail to run Firth logistic regression with the same covariates for case/control traits. As we observed that some mitochondrial DNA variants were specific to array type, we also ran linear regression including array type as a covariate; we did not perform logistic regression with array type as a covariate due to convergence issues from complete separation of

variants assessed only on a single array type. We defined mtDNA-wide significance using a Bonferroni correction by $p = \frac{0.05}{4337} \approx 1.15e - 5$.

## Enrichment analysis of components of organellar proteomes

COMPARTMENTS (RRID:SCR_015561) (https://compartments.jensenlab.org) (*Binder et al., 2014*) is a resource integrating several lines of evidence for protein localization predictions including annotations, text-mining, sequence predictions, and experimental data from the Human Protein Atlas. We used this resource to obtain the degree of evidence (a number ranging from 0 to 5) linking each gene to localization to one of 12 organelles: nucleus, cytosol, cytoskeleton, peroxisome, lysosome, endoplasmic reticulum, Golgi apparatus, plasma membrane, endosome, extracellular space, mitochondrion, and proteasome. To avoid noisy localization assignments due to weak text mining and prediction evidence, we only considered localization assignments with a score > 2 as described previously (*Binder et al., 2014*). We subsequently assigned compartment(s) to each gene by selecting the compartment(s) with the maximal score within each gene. We only included compartments containing over 240 genes due to limited power at smaller gene-set sizes and used MitoCarta2.0 (*Calvo et al., 2016*) to obtain a higher confidence set of genes localizing to the mitochondrion, resulting in gene-sets representing the proteomes of 10 organelles. S-LDSC and MAGMA were used to test for enrichment across the UKB age-related traits for these gene-sets in the usual way, controlling for the set of protein-coding genes. S-LDSC was also used to obtain estimates of the percentage of heritability explained by each organelle gene-set.

## Enrichment analysis of spatial components of the nucleus

To produce interpretable sub-divisions of the nucleus, we used Gene Ontology (GO) (RRID:SCR_017505) (*The Gene Ontology Consortium et al., 2019*; *Ashburner et al., 2000*) to identify terms listed as children of the nucleus cellular component (GO:0005634). We used Ensembl (RRID:SCR_002344) version 99 (*Yates et al., 2020*) to obtain a first pass set of genes annotated to each sub-compartment of the nucleus (or its children). After manual review of sub-compartments with > 90 genes, we selected nucleoplasm (GO:0005654), nuclear chromosome (GO:0000228), nucleolus (GO:0005730), nuclear envelope (GO:0005635), splicosomal complex (GO:0005681), nuclear DNA-directed RNA polymerase complex (GO:0055029), and nuclear pore (GO:0005643). We excluded terms listed as 'part' due to poor interpretability and manually excluded similar terms (e.g. nuclear lumen vs nucleoplasm). To generate a high confidence set of genes localizing to each of these selected sub-compartments, we then turned to the COMPARTMENTS resource which assigns localization confidence scores for each protein to GO cellular component terms. We assigned members of the nuclear proteome to these selected nuclear sub-compartments using same the approach outlined for the organelle analysis (Materials and methods). After filtering our selected sub-compartments to those containing > 240 genes, we obtained four categories: nucleoplasm, nuclear chromosome, nucleolus, and nuclear envelope. The nuclear chromosome annotation was largely overlapping with a manually curated high-quality list of TFs (*Lambert et al., 2018*) however was not exhaustive; as such, we merged these lists to generate the chromosome and TF category. To improve interpretability, we removed genes from nucleoplasm that were also assigned to another nuclear sub-compartment, constructed a list of other nucleus-localizing proteins not captured in these four sub-compartments, and included only genes annotated as localizing to the nucleus (Materials and methods). S-LDSC and MAGMA were used to test for enrichment across the UKB age-related traits for these gene-sets in the usual way while controlling for the set of protein-coding genes (Materials and methods).

## Enrichment analysis of functionally distinct TF subsets

We used a published, curated, high-quality list of TFs (*Lambert et al., 2018*) to partition the Chromosome and TF category into TFs and other chromosomal proteins. To determine which TFs are broadly expressed versus tissue specific, we computed $\tau$ per TF across all selected tissues after removing lowly expressed genes with maximal cross-tissue TPM < 1 (Materials and methods, Appendix 1). The threshold for tissue-specific genes was set at $\tau \geq 0.76$ based on the location of the central nadir of the resultant bimodal distribution (*Figure 3—figure supplement 5A*). To identify terciles of TFs by age, we obtained relative gene age assignments for each gene previously

generated by obtaining the modal earliest ortholog level across several databases mapped to 19 ordered phylostrata (*Litman and Stein, 2019*). DNA-binding domain (DBD) annotations for the TFs were obtained from previous manual curation efforts (*Lambert et al., 2018*). S-LDSC and MAGMA were used to test for enrichment across the UKB age-related traits for these gene-sets in the usual way while controlling for the set of protein-coding genes (Materials and methods). We also tested TFs for enrichment in meta-analyses using S-LDSC and MAGMA with the same parameters as for UKB traits (Appendix 1).

## Analysis of constraint across organelles and sub-organellar gene-sets

We obtained gene-level gnomAD (RRID:SCR_014964) v2.1.1 constraint tables (https://gnomad.broadinstitute.org), haploinsufficient genes, and olfactory receptors (*Karczewski et al., 2020*) (https://github.com/macarthur-lab/gene_lists). Constraint values as loss-of-function observed/expected fraction (LOEUF) were mapped to genes within organelle, sub-mitochondrial, sub-nuclear, and TF binding domain gene-sets.

## Enrichment analysis across age-related disease holding constraint as a covariate

To test for enrichment with constraint as a covariate, we used MAGMA with UKB age-related traits. We mapped variants to genes and performed the gene-level analysis as done previously for the mitochondria-localizing gene and organelle analysis. We included LOEUF and log LOEUF as covariates for the gene-set analysis in addition to the default covariates (gene length, SNP density, inverse MAC, as well as the respective log-transformed versions) via the –condition-residualize flag.

## Acknowledgements

We thank D Altshuler, SE Calvo, T Finkel, H Finucane, ES Lander, ME MacDonald, D Palmer, EB Robinson, AV Segrè, ME Talkowski, RK Walters, CC Winter, and members of the Mootha and Neale labs for critical feedback and discussions. This research has been conducted using the UK Biobank Resource under Application Number 31063. This project was supported in part by grants (NIH R35GM122455 to VKM, NIH R01 MH101244 to BMN, and NIH T32 AG000222 to RG) from the National Institutes of Health.

## Additional information

### Competing interests

Konrad J Karczewski: KJK is a consultant for Vor Biopharma. Benjamin M Neale: BMN is a member of the scientific advisory board at Deep Genomics and RBNC Therapeutics. BMN is a consultant for Camp4 Therapeutics, Takeda Pharmaceutical and Biogen. Vamsi K Mootha: VKM is an advisor to and receives compensation or equity from Janssen Pharmaceuticals, 5am Ventures, and Raze Therapeutics. The other authors declare that no competing interests exist.

### Funding

| Funder | Grant reference number | Author |
| --- | --- | --- |
| National Institutes of Health | T32AG000222 | Rahul Gupta |
| National Institutes of Health | R35GM122455 | Vamsi K Mootha |
| National Institutes of Health | R01MH101244 | Benjamin M Neale |

The funders had no role in study design, data collection and interpretation, or the decision to submit the work for publication.

### Author contributions

Rahul Gupta, Conceptualization, Data curation, Formal analysis, Validation, Investigation, Visualization, Methodology, Writing - original draft, Writing - review and editing; Konrad J Karczewski, Daniel

Howrigan, Methodology, Writing - review and editing; Benjamin M Neale, Conceptualization, Supervision, Methodology, Writing - review and editing; Vamsi K Mootha, Conceptualization, Supervision, Funding acquisition, Methodology, Writing - original draft, Writing - review and editing

### Author ORCIDs
Rahul Gupta (iD) https://orcid.org/0000-0001-8263-2455
Konrad J Karczewski (iD) https://orcid.org/0000-0003-2878-4671
Daniel Howrigan (iD) https://orcid.org/0000-0002-7721-4838
Benjamin M Neale (iD) https://orcid.org/0000-0003-1513-6077
Vamsi K Mootha (iD) https://orcid.org/0000-0001-9924-642X

### Decision letter and Author response
Decision letter https://doi.org/10.7554/eLife.68610.sa1
Author response https://doi.org/10.7554/eLife.68610.sa2

## Additional files
### Supplementary files
• Source data 1. Effect size point estimates and p-values for all tested gene-sets and all tested age-related traits.

• Source data 2. mtDNA GWAS summary statistics.

• Source data 3. Effect size point estimates and p-values for all tested gene-sets for aging phenotypes.

• Supplementary file 1. Tested traits and sample sizes in UK Biobank and external meta-analyses.

• Supplementary file 2. (a) QC calls assigned to mtDNA variants through manual review of cluster plots. (b) Transcription factor assignment to broadly-expressed vs tissue-specific categories using two sub-region groupings.

• Transparent reporting form

### Data availability
Heritability point estimates and standard errors for age-related traits are listed in Supplementary File 1. Genetic and phenotypic correlation point estimates and standard errors/p-values plotted in Figure 1B are available in Figure 1-Source data 1. Summary statistics from mtDNA-GWAS (plotted in Figure 2 and Figure 2—figure supplement 9) are available in Source data 2. All gene-based enrichment analysis p-values and point estimates are available in Source data 1 and Source data 3. Period prevalence data for diseases in the UK can be obtained from Kuan et al. 2019. Gene-sets can be found using COMPARTMENTS (https://compartments.jensenlab.org), MitoCarta 2.0 (https://www.broadinstitute.org/files/shared/metabolism/mitocarta/human.mitocarta2.0.html), Lambert et al. 2018 (DOI: 10.1016/j.cell.2018.01.029), Frazier et al. 2019 (DOI: 10.1074/jbc.R117.809194), Finucane et al. 2018 (https://alkesgroup.broadinstitute.org/LDSCORE/), Kapopoulou et al. 2015 (DOI: 10.1111/evo.12819), and the MacArthur laboratory (https://github.com/macarthur-lab/gene_lists, copy archived at https://archive.softwareheritage.org/swh:1:rev:fcc849637bd71e683bffc618e1a48081a8df08f8). Gene age estimates were obtained from Litman, Stein 2019 (DOI: 10.1053/j.seminoncol.2018.11.002). GWAS catalog annotations can be obtained from: https://www.ebi.ac.uk/gwas. Heritability estimates across UKB can be obtained at: https://nealelab.github.io/UKBB_ldsc/. UKB summary statistics can be obtained from Neale lab GWAS round 2: https://github.com/Nealelab/UK_Biobank_GWAS (copy archived at https://archive.softwareheritage.org/swh:1:rev:dc7b7b590413e-c96a45a64f7213f50a3a0606198). Annotations for the Baseline v1.1 and BaselineLD v2.2 models as well as other relevant reference data, including the 1000G EUR reference panel, can be obtained from https://alkesgroup.broadinstitute.org/LDSCORE/. eQTL and expression data in human tissues can be obtained from GTEx: https://www.gtexportal.org. Constraint estimates can be found via gnomAD: https://gnomad.broadinstitute.org. See citations for publicly available GWAS meta-analysis summary statistics (Teslovich et al., 2010; Ehret et al., 2011; Timmers et al., 2019; Zenin et al., 2019;

Manning et al., 2012; Morris et al., 2012; Schunkert et al., 2011; Estrada et al., 2012; Christophersen et al., 2017; Pattaro et al., 2016; Nalls et al., 2019; Lambert et al., 2013).

The following previously published datasets were used:

| Author(s) | Year | Dataset title | Dataset URL | Database and Identifier |
|---|---|---|---|---|
| Binder JX, Pletscher-Frankild S, Tsafou K, Stolte C, O'Donoghue SI, Schneider R, Jensen LJ | 2014 | COMPARTMENTS | https://compartments.jensenlab.org/Downloads | COMPARTMENTS Portal, COMPARTMENTS |
| Calvo SE, Klauser CR, Mootha VK | 2015 | MitoCarta2.0 | https://www.broadinstitute.org/files/shared/metabolism/mitocarta/human.mitocarta2.0.html | Broad Institute, human.mitocarta2.0 |
| Buniello A, MacArthur JAL, Cerezo M, Harris LW, Hayhurst J, Malangone C, McMahon A, Morales J, Mountjoy E, Sollis E, Suveges D, Vrousgou O, Whetzel PL, Amode R, Guillen JA, Riat HS, Trevanion SJ, Hall P, Junkins H, Flicek P, Burdett T, Hindorff LA, Cunningham F, Parkinson H | 2019 | GWAS Catalog, all associations v1.0.2 | https://www.ebi.ac.uk/gwas/docs/file-downloads | NHGRI-EBI GWAS Catalog, gwas |
| Abbott L, Bryant S, Churchhouse C, Ganna A, Howrigan H, Palmer D, Neale B, Walters R, Carey C, The Hail team | 2018 | Neale Lab UKB Round 2 GWAS Summary Statistics | http://www.neale-lab.is/uk-biobank/ | Neale lab, uk-biobank |
| Walters R, Baya N, Tashman K, Chen D, Abbott L, Carey C, Palmer D, Neale B | 2019 | UKB Round 2 GWAS Heritability Estimates | https://www.dropbox.com/s/8vca84rsslgbsua/ukb31063_h2_topline.02Oct2019.tsv.gz?dl=1 | Dropbox, 8vca84rsslgbsua |
| Teslovich TM | 2010 | Biological, clinical and population relevance of 95 loci for blood lipids | http://csg.sph.umich.edu/willer/public/lipids2010/ | University of Michigan, lipids2010 |
| The International Consortium for Blood Pressure Genome-Wide Association Studies | 2011 | Genetic variants in novel pathways influence blood pressure and cardiovascular disease risk | https://www.ncbi.nlm.nih.gov/projects/gap/cgi-bin/analysis.cgi?study_id=phs000585.v2.p1&phv=&phd=&pha=3588&pht=&phvf=&phdf=&phaf=1&phtf=&dssp=1&consent=&temp=1 | dbGaP phs000585.v1, phs000585.v1 |
| DIAGRAM Consortium | 2012 | Large-scale association analysis provides insights into the genetic | https://www.diagram-consortium.org/downloads.html | DIAGRAM T2D Stage 1 GWAS, 1 GWAS |

| | | | | |
|---|---|---|---|---|
| | | architecture and pathophysiology of type 2 diabetes, stage 1 GWAS | | |
| CARDIoGRAM plus C4D Consortium | 2011 | Large-scale association analysis identifies 13 new susceptibility loci for coronary artery disease | http://www.cardio-gramplusc4d.org/media/cardiogram-plusc4d-consor-tium/data-down-loads/cardiogram_gwas_results.zip | CARDIoGRAM plus C4D meta-analysis, meta-analysis |
| GEnetic Factors for OSteoporosis Consortium | 2012 | Genome-wide meta-analysis identifies 56 bone mineral density loci and reveals 14 loci associated with risk of fracture | http://www.gefos.org/sites/default/files/GEFOS2_FNBMD_POOLED_GC.txt.gz | GEFOS Pooled Femoral Neck Summary Statistics, GEFOS2_FNBMD_POOLED_GC |
| AFGen | 2017 | Large-scale analyses of common and rare variants identify 12 new loci associated with atrial fibrillation | https://personal.broadinstitute.org/ryank/28416818.2017.AFGen.GWAS.zip | Human Genetics Amplifier, 28416818.2017 |
| AFGen | 2016 | Genetic associations at 53 loci highlight cell types and biological pathways relevant for kidney function; eGFRcrea and CKD | https://ckdgen.imbi.uni-freiburg.de/#Pattaro2016da-ta | CKDGen Data at Medical Center - University of Freiburg, Pattaro2016data |
| Brainstorm, IPDGC | 2019 | Identification of novel risk loci, causal insights, and heritable risk for Parkinson's disease: a meta-analysis of genome-wide association studies | https://drive.goo-gle.com/file/d/1FZ9UL99LAqyW-nyNBxxlx6qOUlfA-nublN/view?usp=sharing | IPDGC GWAS META5 summary stats (excluding 23andMe), 1FZ9UL99LAqyWnyNBxxlx6qOUlfAnublN |
| International Genomics of Alzheimer's Project (IGAP) | 2013 | Meta-analysis of 74,046 individuals identifies 11 new susceptibility loci for Alzheimer's disease | https://www.nia-gads.org/datasets/ng00036 | IGAP Stage 1, ng00036 |
| Timmers PRHJ, Mounier N, Lall K, Fischer K, Ning Z, Feng X, Bretherick AD, Clark DW, eQTLGen Consortium, Shen X, Esko T, Kutalik Z, Wilson JF, Joshi PK | 2019 | Genomics of 1 million parent lifespans implicates novel pathways and common diseases and distinguishes survival chances | https://datashare.ed.ac.uk/handle/10283/3209 | Edinburgh DataShare, 10.7488/ds/2463 |
| Zenin A, Tsepilov Y, Sharapov S, Getmantsev E, Menshikov LI, Fedichev PO, Aulchenko Y | 2019 | Identification of 12 genetic loci associated with human healthspan | https://zenodo.org/record/1302861 | Zenodo, 10.5281/zenodo.1302861 |
| GTEx Consortium | 2019 | GTEx v8 median expression TPM | https://storage.goo-gleapis.com/gtex_ | GTEx portal, GTEx_Analysis_2017-06-05_v8_RNASeQCv1.1.9_gene_median_tpm |

| | | | | | |
|---|---|---|---|---|---|
| | | per tissue | analysis_v8/rna_seq_data/GTEx_Analysis_2017-06-05_v8_RNASeQCv1.1.9_gene_median_tpm.gct.gz | | |
| GTEx Consortium | 2019 | GTEx v8 single tissue eQTLs | https://storage.googleapis.com/gtex_analysis_v8/single_tissue_qtl_data/GTEx_Analysis_v8_eQTL.tar | GTEx portal, GTEx_Analysis_v8_eQTL | |

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

# Appendix 1

## Choice of traits with meta-analyses with cohorts separate from UKB

For 10 traits with well-powered UKB GWAS and meta-analyses, we ensured that the meta-analyses used did not incorporate data from UKB thus allowing their use as replication cohorts. Parkinson's Disease (*Nalls et al., 2019*) and Alzheimer's Disease (*Lambert et al., 2013*) were analyzed as part of meta-analyses but not UKB due to power limitations in UKB and eGFR was assessed only in the tested meta-analysis. In the case of Parkinson's Disease, a well-powered GWAS was recently performed and included UKB individuals (*Nalls et al., 2019*). Given that this trait was not sufficiently powered for analysis in UKB alone, we chose to proceed with summary statistics from this study. Because mtDNA-GWAS could only be performed in UKB (where we had access to individual-level data), we were unable to explicitly test for mtDNA associations with Parkinson's disease, Alzheimer's disease, and eGFR.

## Heritability Z-score threshold selection

Total SNP heritability Z-score encapsulates variables such as polygenicity, sample size, and underlying disease heritability, all of which influence S-LDSC power (*Finucane et al., 2015*). Previous work has indicated that genetic correlation estimates from LD score regression are noisy for total SNP heritability Z-score < 4 (*Bulik-Sullivan et al., 2015*), and total SNP heritability Z-score > 7 has been used as a condition for trait inclusion for S-LDSC (*Finucane et al., 2015*). We decided to use a more relaxed cutoff of total SNP heritability Z-score > 4 for two major reasons: First, we used a distinct enrichment methodology, MAGMA, to validate enrichment signatures. To our knowledge, MAGMA does not produce unstable enrichment estimates for traits with moderate heritability Z-score. Second, we also used GWAS data from non-overlapping cohorts, when available, as independent validation for traits tested in UKB. The lower cutoff was sufficient to produce results that largely replicated across methodology and cohort, while allowing for the inclusion of several traits of interest. Further, several traits with heritability Z-score between 4 and 7 show positive control tissue enrichments and substantial enrichment detection power (for example, LDL levels).

## Choice of traits to test in the GWAS Catalog

We searched the GWAS Catalog phenotypes to identify age-related traits. We manually identified 30 phenotypes that matched our 24 age-related traits (*Figure 2—figure supplement 1A*). This list differs from our full list of age-related traits for two reasons: (1) not all 24 age-related traits had a sufficient number of associated genes for analysis, and (2) in several cases, multiple phenotypes listed in the GWAS catalog matched our age-related traits (e.g. 'Cholesterol, total' and 'Total cholesterol levels'); we tested these separately.

## Investigation of mitochondria-relevant transcription factors in the GWAS Catalog

We tested if any of eight TFs known to regulate mitochondrial function – *TFAM, GABPA, GABPB1, ESRRA, YY1, NRF1, PPARGC1A, and PPARGC1B* – were the nearest gene to any genome-wide significant variants listed for age-related traits in the GWAS Catalog. We tested the same traits we used for enrichment analysis of the MitoCarta genes in the GWAS Catalog (*Figure 2—figure supplement 1*) and did not find any signal for 29/30 tested phenotypes. We did find that TFAM was one of the nearest genes for heel bone mineral density, however we note that there are a total of 1496 unique mapped nearest genes for this trait. Further, we tested mitochondria-localizing genes for enrichment in GWAS for heel bone mineral density (3148_irnt) in UKB and found no evidence of enrichment (*Figure 2*).

## Choice of enrichment method

In this study, we leveraged several enrichment methods to ensure robustness to methodology. We used Fisher's exact test in a first-pass analysis of enrichment of GWAS signal in the GWAS Catalog. While this provides a useful preview of the enrichment landscape across published GWAS, this

suffers from numerous limitations, including the usage of only genome-wide significant SNPs, the treatment of each variant as equally likely to contribute to GWAS signal under the null, and an inability to easily control for covariates such a gene length, among others. As such, we used two different methods, MAGMA and S-LDSC, to test for GWAS enrichment among our gene-sets while resolving these confounders and reducing the likelihood of model misspecification. We used S-LDSC to test for heritability enrichment within specified variants controlling for 53 functional categories including DNase hypersensitivity sites, H3K4Me sites, and coding regions. MAGMA uses a variation of Fisher's method to obtain gene-level test statistics and test for gene-set enrichment controlling for LD structure, and when performing the gene-set enrichment testing we controlled for gene length, inverse MAC, and SNP density. We also used tissue-specific enrichments as positive controls to ensure that the methods we used were working properly.

Notably, when running MAGMA on age-associated traitmeta-analyses with a 100 kb window, we were unable to find tissue-specific enrichments (*Figure 2—figure supplement 6B*). Given that S-LDSC, and MAGMA with a 5 kb up and 1.5 kb down window, identified these enrichments in the selected meta-analyses (*Figure 2—figure supplement 3B*, *Figure 2—figure supplement 6A*) and that we observe reduced power for enrichments among meta-analyses relative to UKB (*Figure 2—figure supplement 7H*), we attributed the lack of tissue enrichments using MAGMA at 100 kb in meta-analyses to a lack of power. Indeed, MAGMA with a 100 kb symmetric window was able to identify enrichments in UKB (*Figure 2—figure supplement 5B*). Thus, we did not test any other gene-sets among meta-analyses using MAGMA with a 100 kb window.

## Choice of control genes for gene-based tests

For all gene-based analyses we aimed to perform a competitive analysis, testing if our genes of interest explained more trait heritability than comparable loci elsewhere in the genome. For our positive control tests and power analyses leveraging the set of highest expressed tissue-specific genes, we controlled for the set of genes across which t-statistics were computed (~25,000 genes); namely all genes that had at least four samples in GTEx with one or more counts-per-million (*Finucane et al., 2018*). All of our non-tissue gene-sets (e.g. MitoCarta genes, organelle-localizing genes) were subsets of the set of protein-coding genes, so we controlled for the set of protein-coding genes for these analyses (~19,000 genes). For S-LDSC, this involved including the respective control gene-set annotation atop the baseline model; for MAGMA, this involved defining the gene location file based on the control gene-set such that the space of genes considered was restricted to the genes to be controlled for.

## Power analysis of gene-based tests

To verify the power of S-LDSC and MAGMA in our selected traits, we sub-sampled each of ten positive control tissue-trait pairs. We subsampled the set of tissue-expressed genes for each of the six selected tissues at various gene-set sizes and empirically assessed the number of trials in which significant enrichment was detected, giving us an estimate of power, or $\Pr(reject|alternative)$. All tissue enrichments were originally performed with 2485 genes (Materials and methods); as such we conducted subsampling trials with 1523, 1105, 800, and 350 genes to assess power throughout our study. Because LD score computations are very computationally intensive, we generated 50 random subsamples per gene-set size-tissue pair ensuring that each sample contained a proportional number of genes per chromosome to the original tissue expressed gene-set. We mapped variants to genes and computed LD scores per-chromosome for each annotation (Materials and methods). For each gene-set size and tissue (24 gene-set size-tissue pairs), we generated 1000 sets of LD scores by shuffling LD scores computed per chromosome, effectively generating 1000 random tissue gene-set subsamples for each gene-set size-tissue pair. We subsequently used S-LDSC to test for enrichment for each of the 1000 tissue gene-set subsamples in the aforementioned selected traits for each gene-set size, resulting in 240,000 regressions atop the baseline model as performed for the tissue enrichments in the usual way (Materials and methods). The gene-sets generated for use with S-LDSC (1000 per gene-set size-tissue pair) were also exported for analysis using MAGMA with the same competitive analysis performed for the tissue-enrichment analysis (Materials and methods).

To characterize the power differential between UKB and meta-analyses, we tested the subset of the tissue-trait pairs tested in UKB that showed enrichment in the corresponding meta-analysis with either S-LDSC (*Figure 2—figure supplement 3B*) or MAGMA (*Figure 2—figure supplement 6*). This resulted in an assessment of power among meta-analyses for liver-TG, liver-LDL, liver-HDL, visceral adipose-HDL, atrial appendage-atrial fibrillation, pancreas-T2D, and pancreas-glucose. We tested the same gene-sets tested with S-LDSC and MAGMA (1000 per gene-set size-tissue pair) in UKB using both S-LDSC and MAGMA in the usual way (Materials and methods).

As expected, we noted that power was a function of both enrichment effect size and gene-set size for S-LDSC and MAGMA (*Figure 2—figure supplement 7A*–*Figure 2—figure supplement 7F*). While we observed lower power across most tested traits among meta-analyses when compared to UKB, power was acceptable among the meta-analyses for high effect size enrichments for gene-sets with 1105 genes (*Figure 2—figure supplement 7G*, *Figure 2—figure supplement 7H*).

## Choice of gene-sets to test for replication among meta-analyses

Because our power analyses showed a substantial reduction of power for tested meta-analyses relative to UKB (*Figure 2—figure supplement 7I*, *Figure 2—figure supplement 7J*), we tested only a subset of all tested gene-sets for replication among meta-analyses. Namely, we sought to test replication of the two major organelle-based results in this study: (1) the lack of enrichment of mitochondria-localizing genes across age-related disease and (2) the enrichment of chromosome and TF genes, with subsequent enrichment of the TFs alone.

## Choice of tissues to include for multi-tissue analyses

An assumption key to several statistics used for the eQTL and TF breadth of expression analyses ($\tau_x$, $N_x^{eQTL}, N_x^{express}$) is that different tissues are not overrepresented in the set of tissues assessed. This assumption breaks down in GTEx, where the brain, artery, and esophagus were sampled in multiple sub-regions and the skin, cervix, colon, and adipose tissue were sampled in two sub-regions. We selected specific sub-regions as shown in *Figure 3—figure supplement 5B* manually as expression profiles within sub-regions tend to be far more similar than profiles between sub-regions. To test robustness, we selected an alternate set of tissues within each class (brain frontal cortex (ba9), artery tibial, esophagus muscularis, skin sun exposed (lower leg), colon transverse, and adipose subcutaneous) and repeated our analyses. For our eQTL analysis, we find results that are very similar using this alternate set of tissues as expected (*Figure 2—figure supplement 8C*). Further, we found that with the cutoff of $\tau = 0.76$ for a tissue-specific gene, only 32 of the 1463 tested TFs would be classified differently (*Supplementary file 2b*). Using our original choice of tissues, we find that 605 TFs are tissue specific (*Lambert et al., 2018* report 542 tissue-specific TFs), that 75% of homeodomain containing TFs are tissue specific (*Lambert et al., 2018* report 82%), and that 18.6% of KRAB ZF TFs are tissue specific (*Lambert et al., 2018* report 12%). Thus, our results using our choice of tissues are robust to the specific choice of tissue sub-region within a tissue region and are in good agreement with previously reported tissue-specific expression annotations.

## Model selection for eQTL analyses

To understand if genetic variation near genes localizing to a given organelle were abnormally unlikely to produce downstream biological consequences, we turned to cis-eQTLs. Because most genes have a measured cis-eQTL in at least one tissue (*Figure 2—figure supplement 8A*), we constructed a model to test if genes localizing to a given organelle had significant cis-eQTLs in more or less tissues than other protein-coding genes. We included several covariates to minimize the risk of confounding from first principles (Materials and methods). We corrected for $gene\,length$ and $\log_{10}(gene\,length)$ as we expected that higher number of SNPs in longer genes would increase the probability of eQTL detection; $N_x^{express}$ as we suspected that genes would have detectable eQTLs at most in tissues where they were expressed; and $\tau_x$ as we expected that broadly expressed genes would be more likely to have cis-eQTLs detected in more tissues. Upon model fitting, we observed that all coefficients were significantly different from 0.

## Manual variant QC for mtDNA-GWAS

We used two strategies to manually review the variants that made it through automated variant QC filters (Materials and methods). First, we visually reviewed fluorescence cluster plots for each mtDNA variant to ensure that our variant calls were accurate (Materials and methods). We visually categorized each variant into five categories: clear pass, batch concern, off target variant (OTV) concern, resolution concern, and misclustering (*Supplementary file 2a*), removing 19 variants from further analysis due to cluster plot abnormalities. Second, we computed the mtDNA LD matrix finding no evidence of distance-dependent LD on the mtDNA (*Figure 2—figure supplement 9A*) as observed previously (*Yamamoto et al., 2020*).

## Minor allele frequency filters for mtDNA-GWAS

We used two variant frequency filters to ensure that our regression test statistics were well-behaved (Materials and methods). For continuous traits, we included only variants that had at least 20 individuals with an alternate genotype. For binary traits, we implemented a per-trait and per-variant filter by computing the proportion of individuals with an alternate genotype required such that, under null expectation, there would be at least 20 cases with an alternate genotype. This filter has been shown to eliminate false positive associations by eliminating low MAC variants for rare traits, in which highly imprecise allele frequency estimates can exert high leverage on test statistics (*Howrigan et al., 2017*). This was operationalized as an MAF cutoff as there are by definition no heterozygotes on the mitochondrial DNA, such that for each trait we included only variants that satisfied $MAF \geq 20/\min(CaseSampleSize, ControlSampleSize)$. The sample size estimates were dependent on the variant being assessed as certain variants had distinct missingness patterns due to measurement on a particular genotype array used for only a subset of the cohort (Methods). In total, we tested up to 213 variants per phenotype, assessing a total of 4337 variant-phenotype pairs.

## Enrichment analysis of Parkinson's disease

Of course, much interest lies around characterizing the involvement of mitochondrial dysfunction in PD (*Nguyen et al., 2019*; *Grünewald et al., 2019*; *Abou-Sleiman et al., 2006*; *Ge et al., 2020*). We find no evidence of heritability enrichment among MitoCarta genes in a recent PD GWAS (*Nalls et al., 2019*; *Figure 2D*). Due to power limitations, we were unable to assess mtDNA associations with PD (Appendix 1), though to our knowledge, broadly reproducible associations between inherited mtDNA variants and PD have yet to be reported (*Bose and Beal, 2016*; *Müller-Nedebock et al., 2019*).

## Interpretation of heritability explained by organellar gene-sets

For the sets of genes corresponding to organellar proteomes, we highlight the substantial amount of SNP-heritability explained by variants in or near genes contributing to the nuclear proteome. It is notable that all organelles show $proph^2_{SNP}/propSNP > 1$ (*Figure 3—figure supplement 1*). We believe that this is because of other properties of the SNPs near organelle-localizing genes, namely that all selected SNPs are near protein coding genes. SNPs in protein coding regions are known to be enriched for heritability (*Finucane et al., 2015*), and indeed when we explicitly model these potentially confounding functional SNP annotations (DNase hypersensitivity sites, H3K4Me sites, coding regions; Materials and methods) only the enrichment among variants near nucleus-localizing genes persists.

## Overlap analysis of subsets of the nuclear proteome

We performed pairwise overlap analysis for our five final sub-nuclear compartments (Nucleoplasm, Chromosome and TF, Nucleolus, Nuclear Envelope, Other Nuclear Proteins), finding that virtually all pairs showed an overlap of less than 5% (with an exception for the nucleolus, ~13% of which was also represented in chromosome and TF). S-LDSC and MAGMA were used to test for enrichment across the UKB age-related traits for these gene-sets as performed previously for the organelle analysis.

## GWAS enrichments of functional subdivisions of the class of TFs

We further subdivided the TFs based on breadth of expression in human tissues, DNA-binding domain (DBD), and gene age (Materials and methods). We found a similar pattern of enrichment for tissue-specific TFs and broadly expressed TFs (*Figure 3—figure supplement 5C*, *Figure 3—figure supplement 6A*, *Figure 3—figure supplement 7A*). However, upon stratification by the three largest categories of TF DBD (*Lambert et al., 2018*), we found that non-zinc finger TFs showed enrichment for many age-related traits (*Figure 3—figure supplement 5D*, *Figure 3—figure supplement 6B*, *Figure 3—figure supplement 7B*, *Figure 3—figure supplement 8B*), while the KRAB domain-containing zinc fingers (KRAB ZFs), were largely devoid of enrichment even compared to non-KRAB ZFs (*Figure 3—figure supplement 5E*, *Figure 3—figure supplement 6C*, *Figure 3—figure supplement 7C*, *Figure 3—figure supplement 8C*). While our power analysis suggests sufficient power only for high effect sizes at ~350 genes, we note that (1) the KRAB ZFs and non-KRAB ZFs have similar gene-set sizes and (2) S-LDSC coefficient point estimates are systematically much higher for non-KRAB ZFs than for KRAB ZFs (*Figure 3—figure supplement 7C*). Notably, while we initially observed enrichment only for ancient and intermediate-age TFs but not recently evolved TFs (*Figure 3—figure supplement 5G*, *Figure 3—figure supplement 6D*, *Figure 3—figure supplement 7D*, *Figure 3—figure supplement 8D*), we find that old and recent non-KRAB TFs showed similar enrichment profiles (*Figure 3—figure supplement 5I*, *Figure 3—figure supplement 6E*, *Figure 3—figure supplement 7E*, *Figure 3—figure supplement 8E*), suggesting that the lack of signal among recent TFs was likely attributable to the KRAB domain containing ZFs which are predominantly recently-evolved (*Figure 3—figure supplement 5H*).

## Age-related disease GWAS enrichment with constraint as a covariate

We wanted to assess if our observed enrichment results persist after explicitly accounting for any variance explained by the degree of constraint. We used MAGMA and included LOEUF as a covariate in the gene-set enrichment analysis model (Materials and methods), finding that the LOEUF correction did not substantially impact MitoCarta gene enrichment (*Figure 5—figure supplement 2A*, *Figure 5—figure supplement 3A*) but did reduce the degree of enrichment seen for nucleus-localizing genes (*Figure 5—figure supplement 2B*, *Figure 5—figure supplement 3B*). We continue observing enrichment for the TFs across several age-related diseases (*Figure 5—figure supplement 2E*, *Figure 5—figure supplement 2F*) with a similar pattern of enrichment in non-ZF TFs and non-KRAB ZFs (*Figure 5—figure supplement 2G*) to that seen with the original model (*Figure 3—figure supplement 5D*, *Figure 3—figure supplement 5E*). Thus, while constraint explains a substantial component of the enrichment observed for the TFs among age-related diseases, an enrichment signal persists after accounting for LOEUF.

