## [Decision Letter]

**Acceptance summary:**

The paper provides evidence that genetic variation underlying different age-related diseases mostly influences the functioning of the nucleus, and will be of interest to researchers working on aging.

**Decision letter after peer review:**

Thank you for submitting your article "Human genetic analyses of organelles highlight the nucleus, but not the mitochondrion, in age-related trait heritability" for consideration by *eLife*. Your article has been reviewed by 3 peer reviewers, including Sara Hägg as Reviewing Editor and Reviewer #1, and the evaluation has been overseen by Matt Kaeberlein as the Senior Editor. The following individual involved in review of your submission has agreed to reveal their identity: Joris Deelen (Reviewer #2).

Essential revisions:

1) The rationale for the hypothesis should be better introduced. What is the heritability of these age-related traits that is inferred? What are their genetic correlations? How heritable is mitochondrial dysfunction, are there any such estimates?

Line 74: We hypothesized that heritability for common, age-related traits would be overrepresented among mitochondria-relevant loci, namely variants near genes encoding the organelle's proteome or loci associated with quantitative readouts of mitochondrial function.

2) Please include a reference for the statement or delete:

Line 61: Dysfunction in the mitochondria "……." has been nominated as a driver of virtually all common age- associated diseases.

3) UKB is not yet a reliable source for studying age-related traits because of its age restriction in recruitment (40-70 years). A decline in function with age is normally considered around the age of retirement, which is 65-70 years. For diseases, a similar cut-off is used to differentiate between early-onset (familial) or late-onset cases in e.g., Alzheimer´s disease. Would a different age of onset have given another result?

4) Mitochondrion is a complex organelle that has its own DNA. Compensatory induction of mtDNA copy number and consequently heterogeneity may influence the assessment of mitochondria-relevant variation. Could the mtDNA variants reported in gnomad be potentially confounded by such effects?

5) For age-related disease, there are numerous known mitochondria related genes, such as Parkin and PINK1. Are there any explanations why those genes are missed by the enrichment signals? I may suggest consider also rare age-related disease to see whether there is enrichment.

6) It would be interesting if the authors would also include more direct (endo)phenotypes of ageing, such as parental lifespan (PMID: 30642433) and healthspan (PMID: 30729179) to see if these traits also show enrichment of genetic variation relevant for the functioning of the nucleus.

7) Define what "haplo(in)sufficient" is.

8) There are some errors in the figure legend and text. For example, line 153 the figures seem to be Fig2D and S10; Line158 the figure should be S10E; Line180 the figure should be S11; FigureS8B should link to figure2E not 2D, etc. In the text, when talking about mtDNA loci, it says 213 common variants passing quality control but in figure2A and S10 showed 217. I recommend the authors double-check the whole manuscript for consistency.

9) There is an issue with the quality of Supplementary Table 1, which makes it very hard to read.

10) T2D is abbreviated, use it. TF, GERD is not defined.*Reviewer #1:*

Gupta and co-authors have investigated organelle gene enrichments in age-related diseases using the large-scale UK Biobank cohort data. They hypothesize that as mitochondrial dysfunction is a hallmark of aging, common gene variants linked to the function of the mitochondria should also be linked to different age-related diseases. The authors use state-of-the art methods to investigate complex genetic associations in the hitherto largest prospective cohort available, which would be of interest to researchers in the field of human genetics.

*Reviewer #2:*

The manuscript by Gupta et al., reports the results from a study in which the authors tried to assess the involvement of genetic variation underlying age-related diseases in the functioning of different cellular organelles. They started with the mitochondria, given its well-known role in ageing, but were unable to find enrichment of genetic variation underlying age-related diseases in loci relevant for the functioning of this organelle. They then decided to focus on the remaining cellular organelles and found that the nucleus is the only organelle for which they observed enrichment of genetic variation underlying multiple age-related diseases. They subsequently show that (non-KRAB domain-containing) transcription factors seem to be the main driver of this enrichment. Last but not least, they used data from gnomAD to show that genes encoding the nucleus, and more specifically transcription factors, have a low tolerance to predicted loss-of-function variation (i.e. they are "haploinsufficient").

The major strength of this study is that the authors used several different methods to rigorously analyse different (publicly available) datasets to make sure their findings are robust. I was unable to detect major flaws in the study and think the key claims made by the authors are well supported by the provided data. I was impressed by the amount of data the authors provided to support their 'negative' findings for the mitochondria and really enjoyed reading the manuscript.

The findings show that the age-related decline in integrity of most organelles is likely not due to genetic variation in genes encoding these organelles. Hence, future studies should thus try to identify the mechanisms by which genetic variation in genes encoding transcription factors can contribute to dysfunction in other organelles with age.*Reviewer #3:*

In this manuscript, the authors systematically analyzed the association between inherited genetic variation impacting organelles and assessed their relevance for age-related human diseases. They selected 24 age-related traits and focused on the organelle of mitochondria. Against common expectation, they found no convincing evidence of enrichment for common age-associated diseases among mitochondria-relevant loci. They further tested nine other organelles and found that only the nucleus showed enrichment among many age-associated traits, with the signal emanating from the transcription factors. Fitness analysis also showed nucleus proteome is more constrained than mitochondrial proteome. Given these evidences, the authors concluded that common variants influencing nuclear genome regulation were more related to age-associated diseases than variants influencing individual organelles. In summary, it is an interesting and comprehensive study but the conclusion is against common sense especially those regarding mitochondria as was also noticed by the authors themselves.

Major strengths:

– It is interesting to explore the genetic contribution of organelles in common age-related disease. It is a relatively comprehensive study of mitochondria-relevant variation in age-related disease. As the results were not as expected, they used two robust methods and independent datasets to confirm their findings.

– Further trancing the source of the enrichment signal of nucleus proteome implies the importance of transcription factors. However as TFs are centrally connected components in the proteomic network, which impacts all other genes, it is no surprise a result at all. Could there be any other insights we might draw from this observation?

– It is the first research that systematically evaluated heterogeneity in average pLoF across cellular organelles.

Weakness:

– They only considered common age-related diseases and common genetic variants. The current GWAS enrichment method could be inherently limited for comparing organelles, which was also noticed by the authors themselves in the discussion. I therefore believe the conclusions regarding genetic contribution of different organelle is a little overstated.

– The relationship between enrichment and constraint is a little confusing. Variants associated with age-related diseases are expected to be under weaker selective pressure than early onset diseases. It will be interesting to further explore the constraint results.

---

## [Author Response]

Essential revisions:1) The rationale for the hypothesis should be better introduced. What is the heritability of these age-related traits that is inferred? What are their genetic correlations? How heritable is mitochondrial dysfunction, are there any such estimates?Line 74: We hypothesized that heritability for common, age-related traits would be overrepresented among mitochondria-relevant loci, namely variants near genes encoding the organelle's proteome or loci associated with quantitative readouts of mitochondrial function.

Thank you for your comment. We have modified the second sentence of the introduction to more explicitly list heritabilities of age-related traits. Further, we now include empirical estimates of SNP-heritability for all age-related traits analyzed in Supplementary File 1. We note that we discuss the genetic correlation landscape among age-related traits in the first Results section as part of Figure 1B.

We have also added the following sentence to the paragraph containing the quote above to introduce current heritability estimates of mitochondrial dysfunction:

This genetic approach is supported by the observation that heritability estimates of measures of mitochondrial function are substantial (33-65%^24,25^).

We do note, however, that to our knowledge a large-scale, well-powered assessment of the heritability of other measures of mitochondrial dysfunction has yet to be performed. mtCN is only one of several markers of mitochondrial “function.” As we now mention in the discussion, efforts to develop novel, reliable measures of mitochondrial function and dysfunction may help address this open question.

2) Please include a reference for the statement or delete:Line 61: Dysfunction in the mitochondria "……." has been nominated as a driver of virtually all common age- associated diseases.

The reviewers are correct, we have revised this statement to read “Dysfunction in the mitochondria … has been observed in many common age-associated diseases”. We have also added several citations to support this revised statement, specifically: Lane et al., 2015, Petersen et al., 2004, Mootha et al., 2003, Schapira et al., 1990, Bender et al., 2006, Wanagat et al., 2001, and Ashar et al., 2017.

3) UKB is not yet a reliable source for studying age-related traits because of its age restriction in recruitment (40-70 years). A decline in function with age is normally considered around the age of retirement, which is 65-70 years. For diseases, a similar cut-off is used to differentiate between early-onset (familial) or late-onset cases in e.g., Alzheimer´s disease. Would a different age of onset have given another result?

Thank you for your comment. We specifically leverage epidemiological data from the UK (Kuan et al., 2019 Lancet Digital Health) to select common age-related diseases prior to our analysis in UKB to avoid biasing our disease selection process to the UKB. These data are plotted in Figure 1A, which shows that our selected traits all have increasing period prevalence with age in the UK even past 60-69 years. Indeed, the only traits we assess with median age-of-onset 50-59 years are GERD, deafness, and abnormal blood pressure / hyperlipidemia traits, which all have very high prevalence in older cohorts (Figure 1A). The majority of our traits would thus be selected even if we restricted to traits with median age-of-onset of 60 years or greater.

To alleviate concerns that our observations are cohort-specific, we replicate our enrichment results with published meta-analyses. Specifically, we analyzed summary statistics from GWAS for atrial fibrillation, CAD, diastolic BP, systolic BP, glucose, HDL, LDL, TG, T2D, and BMD which did not incorporate UKB, providing a replication cohort not subject to concerns related to UKB cohort composition. Our topline findings were replicated in these data – mitochondrial genes showed no enrichment (Figure 2D), and we replicated enrichment among TFs for several of these phenotypes (Figure 3—figure supplement 4) despite worse power (Figure 2—figure supplement 6). Because AD and PD had very low case counts in UKB, we relied on well-powered external meta-analyses of these traits, observing no mitochondrial enrichment for either phenotype (Figure 2D) and some enrichment for TFs in AD (Figure 3—figure supplement 4). Our constraint analysis was performed using gnomAD, which also sources genomic data from a wide variety of cohorts.

We respectfully disagree that UKB is thus far unreliable for studying the genetics of age-related disease. While we agree that the UKB study population is likely not a representative sample of the general population, individuals 60-69 years old are very well represented (https://biobank.ndph.ox.ac.uk/showcase/field.cgi?id=21022). As shown in Supplementary File 1, 19/21 traits assessed in UKB have average ages of onset within the represented ages in UKB. Further, our focus on genetics allows us to avoid bias due to contributors to phenotypic variance in this cohort that have little contribution from genetics. In recent years, UKB has been used with great success to discover or replicate genomic associations for (to name a few): T2D (Xue et al., 2018 Nat Comm), CVD (Klarin et al., 2017 Nat Genet), blood pressure (Evangelou et al., 2018 Nat Genet), cholesterol, HDL and LDL (Sinnott-Armstrong et al., 2021 Nat Genet), osteoporosis (Morris et al., 2019 Nat Genet), GERD (An et al., 2019 Nat Comm), and atrial fibrillation (Roselli et al., 2018 Nat Genet).

4) Mitochondrion is a complex organelle that has its own DNA. Compensatory induction of mtDNA copy number and consequently heterogeneity may influence the assessment of mitochondria-relevant variation. Could the mtDNA variants reported in gnomad be potentially confounded by such effects?

We don’t understand the reference to gnomAD, as our analysis of mtDNA was performed in UKB and our constraint analysis (which did reference gnomAD) was centered on nucDNA-encoded genes only. Aside from our mtDNA-GWAS, the rest of our study involves analyses conducted on the nucDNA.

In the context of our UKB mtDNA-GWAS, we believe that the stated concern is unlikely. The current study is not investigating “heteroplasmic variants,” rather we are investigating the haplotype-defining, inherited mtDNA variants. We explicitly mention in our Discussion that this study is not focused on somatic mutations and rather is concerned with inherited variation.

In our study, we manually inspected fluorescence cluster plots for all UKB samples for each of the 265 variants genotyped on the UKB array to exclude variants that show fluorescence contrast or intensity abnormalities. More specifically, we expected homoplasmic variants to show two well-defined and well-separated clusters and excluded any variants that showed signals in-between two well-separated clusters or that did not show two well-separated clusters. We have described this QC approach in more detail in the Methods and Appendix (under Manual variant QC for mtDNA-GWAS), and we specify the QC outcomes for each of the UKB genotyped SNPs in Supplementary File 2a. Further, we explicitly exclude rare mtDNA variants by imposing minor-allele count cutoffs – including only variants with alternate individual count > 20 for continuous traits and expected minor individual case count > 20 for dichotomous traits. See Appendix for more details (under Minor allele frequency filters for mtDNA-GWAS).

5) For age-related disease, there are numerous known mitochondria related genes, such as Parkin and PINK1. Are there any explanations why those genes are missed by the enrichment signals? I may suggest consider also rare age-related disease to see whether there is enrichment.

Thank you for this important comment. In our study we do not argue against the notion that there may lie associations harbored within mitochondrial genes and age-related traits – indeed these certainly exist. We show instead by testing for enrichment that these associations are found no more frequently than observed elsewhere in the genome. We have modified the discussion to make this more explicit now:

“Here, we focus on enrichment to place the complex genetic architectures of age-related traits in a broader biological context and prioritize pathways for follow-up. For these highly polygenic traits, any large fraction of the genome may explain a statistically significant amount of disease heritability^61,62^, and indeed associations between individual organelle-relevant loci and certain common diseases have been identified previously^63,64^. For example, variants in the endoplasmic reticular genes WFS1 and ATF6B and the mitochondrial gene ATP5G1 have been associated with common T2D^65^. These genes are present in the respective organelle gene-sets, however unlike TFs, neither the endoplasmic reticulum nor the mitochondrion showed enrichment for T2D.”

Regarding the specific example highlighted, as you have noted, there are several known familial, autosomal recessive forms of PD due to mutations in mitochondria-localized proteins such as PINK1, PRKN, and DJ-1. The most recent PD GWAS (Nalls et al. 2019, Lancet Neurol), which analyzed over 35,000 cases, over 18,000 proxy-cases via UKB, and 1.4 million controls, did not highlight SNPs near any of these genes as associated sporadic PD. Manual analysis of the available summary statistics from this GWAS showed no genome-wide significant SNPs in the vicinities of these genes. Though it remains possible that better powered PD GWAS may reveal associations at these loci, the current lack of observed association near PINK1, PRKN, and DJ-1 in sporadic PD GWAS reinforces why we do not observe enrichment in this context.

6) It would be interesting if the authors would also include more direct (endo)phenotypes of ageing, such as parental lifespan (PMID: 30642433) and healthspan (PMID: 30729179) to see if these traits also show enrichment of genetic variation relevant for the functioning of the nucleus.

Thank you for this interesting suggestion. We have obtained the relevant summary statistics from these GWAS, estimated heritability (via S-LDSC atop the baselineLD 2.2 model, the same approach as used throughout the paper), and computed enrichments for organelle and sub-nuclear gene-sets using MAGMA and S-LDSC. Our results are now presented in a new Figure 4 and Figure 4—figure supplement 1, and we have added a relevant short section to our results describing our findings. In summary we see consistent results to our previous analyses, with enrichment among TFs and nuclear proteins and no evidence of enrichment in mitochondrial genes.

7) Define what "haplo(in)sufficient" is.

Thank you for this comment. To add clarity, we now explicitly define both terms in the last paragraph of the introduction.

8) There are some errors in the figure legend and text. For example, line 153 the figures seem to be Fig2D and S10; Line158 the figure should be S10E; Line180 the figure should be S11; FigureS8B should link to figure2E not 2D, etc. In the text, when talking about mtDNA loci, it says 213 common variants passing quality control but in figure2A and S10 showed 217. I recommend the authors double-check the whole manuscript for consistency.

We have now done so – all of these issues should be resolved.

9) There is an issue with the quality of Supplementary Table 1, which makes it very hard to read.

Thank you for your comment. We believe this should now be resolved.

10) T2D is abbreviated, use it. TF, GERD is not defined.

Thank you for this comment. We have updated the manuscript accordingly. We defined the abbreviations for gastro-esophageal reflux disease in line 105 and transcription factors in line 82.